# Competition for the nascent leading strand shapes the requirements for PCNA loading in the replisome

Emma E Fletcher, Morgan L Jones ⬥ & Joseph T P Yeeles ⬥ ✉

## Abstract

During DNA replication, the DNA polymerases Pol δ and Pol ε utilise the ring-shaped sliding clamp PCNA to enhance their processivity. PCNA loading onto DNA is accomplished by the clamp loaders RFC and Ctf18-RFC, which function primarily on the lagging and the leading strand, respectively. RFC activity is essential for lagging-strand replication by Pol δ, but it is unclear why Ctf18-RFC is required for leading-strand PCNA loading and why RFC cannot fulfil this function. Here, we show that RFC cannot load PCNA once Pol ε has been incorporated into the budding yeast replisome and commenced leading-strand synthesis, and this state is maintained during replisome progression. By contrast, we find that Ctf18-RFC is uniquely equipped to load PCNA onto the leading strand and show that this activity requires a direct interaction between Ctf18 and the CMG (Cdc45-MCM-GINS) helicase. Our work uncovers a mechanistic basis for why replisomes require a dedicated leading-strand clamp loader.

**Keywords** Replisome; Ctf18-RFC; Clamp Loader; PCNA Loading; CMG Helicase
**Subject Categories** DNA Replication, Recombination & Repair; Structural Biology

## Introduction

For processive DNA synthesis in all domains of life, replicative DNA polymerases utilise toroidal sliding clamp processivity factors that are topologically bound around duplex DNA to help maintain their association with the template (Hedglin et al, 2013). The homotrimeric eukaryotic sliding clamp PCNA is essential for DNA replication and also serves as a recruitment platform for many additional DNA processing factors, including proteins involved in sister chromatid cohesion, lagging-strand maturation, chromatin assembly, and DNA repair (Boehm et al, 2016; Choe and Moldovan, 2017). Topological loading of PCNA onto duplex DNA requires the activity of dedicated ATP-dependent clamp loader complexes to break the clamp open and load it around DNA at dsDNA-ssDNA junctions (Gaubitz et al, 2022; He et al, 2024; Hedglin et al, 2013; Schrecker et al, 2022). In eukaryotes, including

human and the budding yeast *S. cerevisiae*, there are two such complexes that share four common AAA+ ATPase subunits named Rfc2-5. A fifth AAA+ ATPase subunit, of which Rfc1 is the archetypal member of four paralogs, is required to complete the functional ATPase module necessary for clamp loading and also gives each clamp loader a unique specificity. Clamp loaders formed from Rfc2-5 and either Rfc1 or Ctf18 function to load PCNA at replication forks (Bambara et al, 1997; Lengronne et al, 2006; Liu et al, 2020). Rfc2-5 also associate with Elg1 (ATAD5 in human) in an RFC-like complex responsible for PCNA unloading, and Rad24 (RAD17 in human) to form a complex that is specialised for loading the heterotrimeric 9-1-1 sliding clamp for checkpoint activation (Kupiec, 2016).

Data from strand-specific chromatin immunoprecipitation (ChIP) experiments have demonstrated that PCNA loading by both Rfc1-RFC (hereafter referred to as RFC) and Ctf18-RFC contributes to PCNA occupancy at active replication forks (Liu et al, 2020). The primary function of RFC is to load PCNA onto the lagging strand for DNA synthesis by Pol δ after the priming of each new Okazaki fragment by Pol α-primase, which occurs every ~250 bp (Garg and Burgers, 2005; Maga et al, 2000; Mossi et al, 2000; Smith and Whitehouse, 2012). Consistent with this role, PCNA occupancy is biased towards the leading strand upon depletion of Rfc1 using an auxin-inducible degron (RFC is an essential protein complex in *S. cerevisiae*) (Liu et al, 2020). Conversely, PCNA levels are reduced on the leading strand in cells lacking Ctf18-RFC (Lengronne et al, 2006; Liu et al, 2020). Thus, there is a division of labour for clamp loading activity at replication forks; RFC-dependent PCNA loading is focused on the lagging strand, while Ctf18-RFC activity is targeted to the leading strand. This model is further substantiated by ChIP-seq data showing that Rfc1 and Ctf18 distribute to nascent lagging and leading strands, respectively (Liu et al, 2020).

The mechanisms that target RFC and Ctf18-RFC primarily to lagging and leading strands, respectively, are only partially resolved. Structures of the core replisome indicate that lagging- and leading-strand synthesis are spatially separated (Georgescu et al, 2017; Jones et al, 2021; Sun et al, 2015). The replisome is organised around the 11-subunit CMG (Cdc45-MCM-GINS) helicase that unwinds DNA via a strand exclusion mechanism (Fu et al, 2011). CMG translocates in an N-tier first orientation on the leading-strand template with 3′ to 5′ directionality (Douglas et al, 2018; Georgescu et al, 2017), pulling the ssDNA through a pore in the centre of a ring formed by the 6 MCM subunits (Mcm2-7). The unwound leading-strand template emerges from the C-terminal

MRC Laboratory of Molecular Biology, Cambridge, UK. ✉E-mail: Jyeeles@mrc-lmb.cam.ac.uk

side of the MCM pore, while the lagging-strand template is excluded from the pore and traverses the N-terminal domains of Mcm3 and Mcm5 where it is engaged by Pol α-primase to initiate primer synthesis (Eickhoff et al, 2019; Jenkyn-Bedford et al, 2021; Jones et al, 2023). Based on this arrangement, lagging-strand synthesis is performed towards the "front" of the replisome with respect to replication fork directionality, and leading-strand synthesis towards the "rear".

RFC has a high affinity for primer-template junctions (Gomes and Burgers, 2001; Hingorani and Coman, 2002) and this property is thought to underlie its ability to rapidly associate with, and load PCNA onto, newly primed lagging-strand DNA. Indeed, such is the affinity of RFC for the 3′ ends of dsDNA–ssDNA junctions it can inhibit primer extension by Pol ε, even in the presence of PCNA (Schauer and O'Donnell, 2017). By contrast, the binding of Ctf18-RFC to primer-template junctions is salt sensitive and inhibited by the single-strand DNA binding protein RPA (Bermudez et al, 2003; Bylund and Burgers, 2005). These limitations can be overcome by the direct binding of Ctf18-RFC to Pol ε that is mediated by two additional subunits of the Ctf18-RFC complex, Ctf8 and Dcc1 (Fujisawa et al, 2017). Together with the C-terminus of Ctf18, Ctf8 and Dcc1 form the Ctf18-1-8 module that binds directly to the catalytic domain of Pol2 (Pol2$_{cat}$), the largest subunit of Pol ε (Garcia-Rodriguez et al, 2015; Murakami et al, 2010; Okimoto et al, 2016; Stokes et al, 2020). Very recent cryo-EM structures of Ctf18-RFC in complex with Pol2$_{cat}$, PCNA, and primer-template DNA revealed additional protein:protein interactions between Pol2$_{cat}$ and Ctf18 that enhance PCNA loading by Ctf18-RFC (Yuan et al, 2024). Moreover, data from the same study indicates that binding of the Ctf18-1-8 module to Pol2$_{cat}$ weakens its association with DNA, which could aid the transfer of DNA from the polymerase to Ctf18-RFC during a putative PCNA loading cycle. Consistent with the Ctf18-1-8:Pol2$_{cat}$ interaction being a key determinant of Ctf18-RFC replisome association and PCNA loading, mutations targeting the interface compromise Ctf18-RFC and PCNA levels on nascent DNA, and Ctf18-RFC localisation with CMG (Liu et al, 2020; Stokes et al, 2020). However, PCNA levels are still higher than in *ctf18Δ* cells and sister chromatid cohesion is largely unaffected, indicating that other mechanisms might exist to target Ctf18-RFC to replication forks (Garcia-Rodriguez et al, 2015; Liu et al, 2020; Stokes et al, 2020).

The mechanistic basis for why the replisome requires Ctf18-RFC as a dedicated leading-strand clamp loader, and why RFC is unable to fulfil this role, has not been fully established. Reconstituted budding yeast replisomes assembled via a regulated pathway that mirrors in vivo replisome assembly perform rapid and efficient PCNA-dependent leading-strand synthesis with RFC as the only clamp loader (Yeeles et al, 2017). Ctf18-RFC is therefore not required for robust leading-strand DNA replication by *S. cerevisiae* replisomes, which is consistent with it not being an essential protein, and therefore its potential contribution to leading-strand replication in vitro has not been carefully examined. In contrast, by reconstituting functional human replisomes via a non-canonical pathway starting from purified CMG, we found that CTF18-RFC markedly increased the rate of leading-strand DNA replication, even under conditions where RFC could load PCNA onto the leading strand, indicating that CTF18-RFC directly participates in leading-strand DNA replication, at least under certain conditions (Baris et al, 2022). Whether Ctf18-RFC can also contribute directly to leading-strand DNA replication in the budding yeast replisome

is unclear, nor is it understood why Ctf18-RFC is required for leading-strand PCNA loading in yeast cells that contain functional RFC, and if additional interactions other than those between the Ctf18-1-8 module and Pol2$_{cat}$ are required for leading-strand PCNA loading. In this work we have used a combination of in vitro DNA replication assays with purified budding yeast proteins, cryo-EM analysis of replisomes containing Ctf18-RFC, and targeted protein:protein interaction screening with AlphaFold 3 to reveal how Ctf18-RFC contributes to leading-strand DNA replication, a mechanistic basis for why Ctf18-RFC is required as a dedicated leading-strand PCNA loader, and unique features that enable it to fulfil this role.

## Results

### Ctf18-RFC accelerates replication in the regulated system

Functional budding yeast replisomes have been reconstituted in vitro from purified proteins via two distinct assembly pathways. The first, which we will refer to as the regulated system, involves the assembly and activation of the CMG helicase, around which the replisome forms, via a regulated pathway starting from MCM double hexamers that are loaded around dsDNA at replication origins (Yeeles et al, 2015; Yeeles et al, 2017). In this system and in vivo, leading-strand DNA synthesis is established when the first Okazaki fragments, that are primed by Pol α-primase and extended by Pol δ, are "handed over" to the CMG-Pol ε (CMGE) complex (Aria and Yeeles, 2018; Garbacz et al, 2018; Zhou et al, 2019). The second approach exploits the fact that functional CMG complexes can be assembled following overexpression of all 11 subunits in *S. cerevisiae* (Georgescu et al, 2014). Because CMG is presumably assembled off DNA following overexpression, it must then be loaded onto a single-stranded leading-strand template at a model replication fork. Replisomes that perform leading- and lagging-strand DNA replication can then be made by the addition of DNA polymerases and replisome accessory factors (Georgescu et al, 2014; Georgescu et al, 2015). We refer to this approach as the CMG-based system.

To examine if Ctf18-RFC could influence leading-strand DNA replication, we purified budding yeast Ctf18-RFC (Fig. EV1A) and first tested its ability to support primer extension by Pol ε and Pol δ. The addition of Ctf18-RFC stimulated primer extension on M13 ssDNA by Pol ε, indicating that the complex was functional for PCNA loading (Fig. EV1B,C). In contrast, Ctf18-RFC could not support DNA synthesis by Pol δ in the presence of RPA and replication was only barely detectable when RPA was substituted for *E. coli* SSB (Fig. EV1D), indicating that Ctf18-RFC could not efficiently load PCNA under these reaction conditions.

Having established that Ctf18-RFC was functional with Pol ε, we performed a DNA replication reaction using the regulated system (Fig. 1A) in the presence of RFC, with Ctf18-RFC either added or omitted. Figure 1B–D shows that the addition of Ctf18-RFC stimulated the maximal rate of leading-strand DNA replication. The increase in rate was ~14% which is considerably less than that previously observed in the human CMG-based system (~40% increase) (Baris et al, 2022). Nevertheless, the data show that Ctf18-RFC influences leading-strand synthesis in the regulated system, even when RFC is present. Furthermore, disrupting Ctf18-RFC ATPase

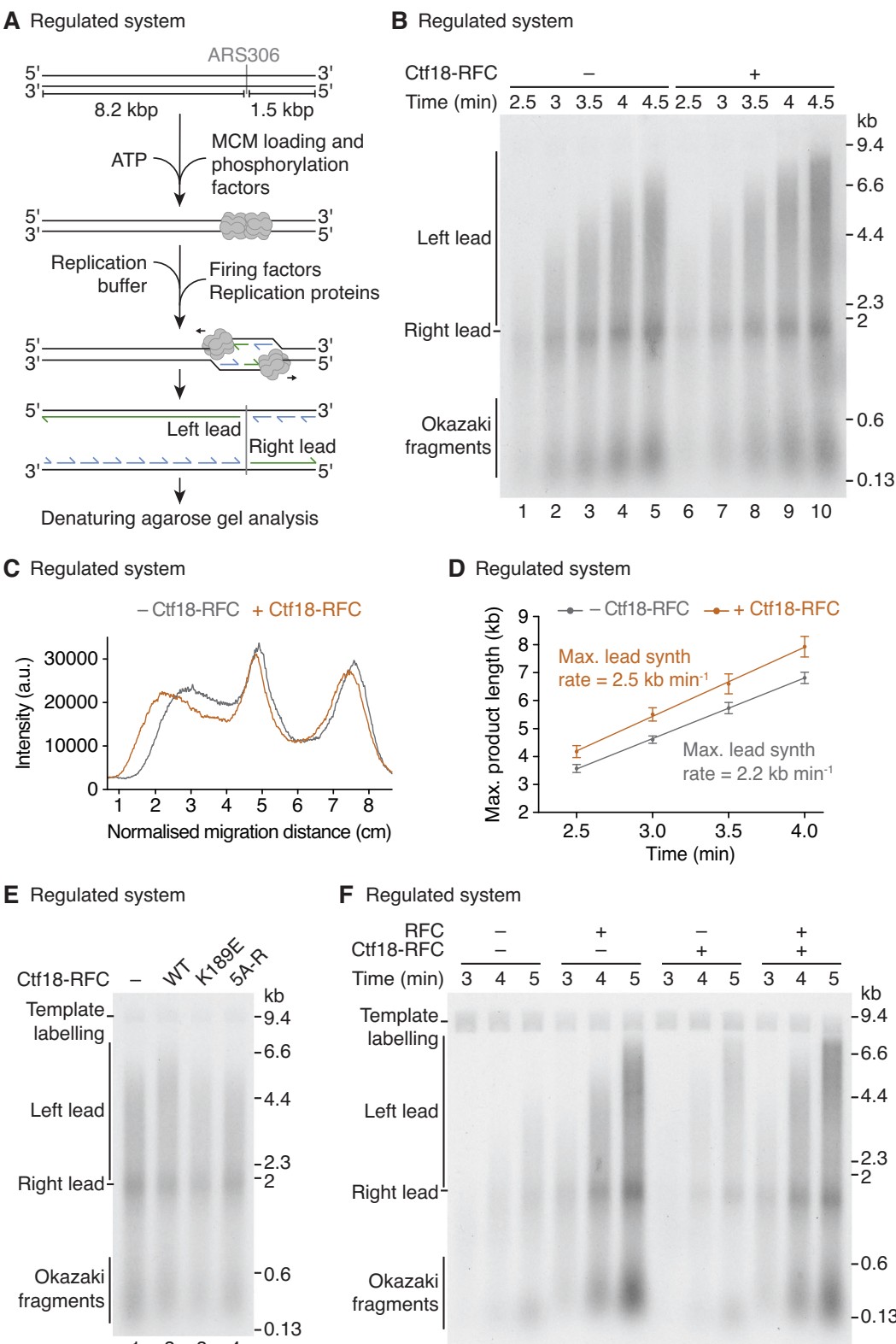

**Figure 1.  Ctf18-RFC accelerates replication in the regulated system.**

(A) Schematic of the regulated in vitro DNA replication system. (B) Denaturing agarose gel analysis of replication reactions using the regulated system as in (A) in the absence or presence of Ctf18-RFC, with RFC present throughout. (C) Lane profiles of the 4 min timepoints from reactions in the absence and presence of Ctf18-RFC as in (B). (D) Quantification of maximal leading-strand synthesis rates from replication reactions in the regulated system as in (B). Linear regression is fit to the mean of three experiments. The error bars represent the standard error of the mean (s.e.m.) and the mean is indicated by filled circles. (E) Denaturing agarose gel analysis of replication reactions using the regulated system with wild-type (WT) or mutant Ctf18-RFC complexes present where indicated. Reactions were analysed after 4 min. In this and subsequent figures, CMG-independent template labelling is indicated. (F) Denaturing agarose gel analysis of replication reactions using the regulated system in the absence or presence of RFC and Ctf18-RFC. Source data are available online for this figure.

activity (Ctf18$^{K189E}$-RFC), or the interaction between the Ctf18-1-8 module and the Pol ε catalytic domain (Ctf18-RFC$^{5A-R}$) (Stokes et al, 2020), prevented replication acceleration (Figs. 1E and EV2A), strongly suggesting that PCNA loading by Ctf18-RFC is required for this function.

To further explore the division of labour between RFC and Ctf18-RFC during leading-strand replication we performed a regulated system reaction with either no clamp loader, RFC, Ctf18-RFC, or both clamp loaders (Fig. 1F). Consistent with previous observations (Yeeles et al, 2017), in the absence of a clamp loader replication was established and leading strands approached ~5 kb after 5 min. In the presence of RFC, there was an increase in both the abundance and length of leading-strand products, indicating that PCNA loading by RFC contributed to both the establishment and rate of leading-strand replication. In contrast, when Ctf18-RFC was the only clamp loader in the reaction, the rate, but not the abundance, of leading-strand products was enhanced (Figs. 1F, compare lanes 9 and 3 and EV2B). Maximal rate and efficiency required both RFC and Ctf18-RFC. Collectively, these results show that in the regulated system RFC is required to promote the efficient establishment of leading-strand synthesis, whereas Ctf18-RFC is necessary for maximal rates.

## Ctf18-RFC, but not RFC, supports PCNA loading at preassembled CMGE

Because the acceleration of leading-strand synthesis by yeast Ctf18-RFC in the regulated system was markedly less than human CTF18-RFC in the CMG-based system (Baris et al, 2022), we decided to examine Ctf18-RFC function in a yeast CMG-based system to determine if the differences reflected species-to-species mechanistic variation or differences in the experimental setup. To do so, purified budding yeast CMG was loaded onto a 9.7 kbp linear template with a model replication fork ligated at one end (Fig. 2A). After addition of the replication proteins Ctf4, Tof1-Csm3, Mrc1, and Pol ε to form replisomes, template unwinding and leading-strand replication were initiated by addition of ATP and RPA. Pol α-primase and Pol δ were omitted so that only leading-strand products could be synthesised. Surprisingly, the addition of RFC did not substantially enhance the rate of leading-strand synthesis, even in the presence of PCNA, and products of only ~4 kb were synthesised by 3 min (Fig. 2B, compare lanes 5 and 6). In reactions with Ctf18-RFC, products approaching the unit length of 9.7 kb were synthesised and this stimulation was strictly PCNA-dependent (Fig. 2B, compare lanes 7 and 3). Moreover, Ctf18-RFC complexes defective in ATPase activity (Ctf18$^{K189E}$-RFC) or Pol ε binding (Ctf18-RFC$^{5A-R}$) failed to enhance leading-strand replication (Fig. EV2C), and mutation of the Pol ε PCNA interacting motif (PIP box) (Aria and Yeeles, 2018) greatly diminished the ability of Ctf18-RFC to stimulate synthesis (Fig. EV2D). These data indicate that Ctf18-RFC stimulates replication by loading PCNA for leading-strand synthesis. Strikingly, primer extension

reactions with Pol ε performed in a similar ionic strength buffer displayed the opposite behaviour, with DNA synthesis almost entirely dependent on RFC-dependent PCNA loading and Ctf18-RFC unable to support DNA synthesis (Fig. 2C). These results emphasise that RFC and Ctf18-RFC display different functionality when operating in primer extension and CMG-based replication reactions.

## Transient uncoupling promotes PCNA loading by RFC

To further assess how Ctf18-RFC was promoting leading-strand synthesis in the CMG-based system we performed a pulse-chase experiment where clamp loaders were added either in the pulse or the chase. Addition in the pulse step reports on effects occurring during or close to the establishment of replication, whereas addition in the chase should reveal effects at established replication forks. Figure 3A shows that Ctf18-RFC stimulated leading-strand synthesis when added either in the pulse or the chase, demonstrating that it can function at established replication forks and potentially also prior to the commencement of leading-strand synthesis. Consistent with the data in Fig. 2B, RFC addition in the chase had minimal, if any, effect on replication. However, RFC addition in the pulse resulted in the synthesis of longer leading-strand products, although they remained shorter than those synthesised in the comparable reaction containing Ctf18-RFC. The addition of Ctf18-RFC in the chase when RFC was included in the pulse further stimulated leading-strand synthesis.

We hypothesised that the ability of RFC to stimulate replication when added in the pulse but not the chase, and not during regular CMG-based reactions, could be due to the altered reaction conditions of the pulse step, specifically a reduction in dCTP concentration from 30 μM to 3 μM. Reducing dNTP concentration has been shown to induce helicase-polymerase uncoupling (Devbhandari and Remus, 2020), which we considered might promote RFC-dependent PCNA loading by destabilising the association of Pol ε with the 3′ end of the nascent leading strand so that RFC could gain access. The suggestion that Pol ε might prevent RFC from accessing the 3′ end is supported by data showing that synthesis by Pol ε in the context of CMGE is refractory to competition by RFC, whereas RFC can inhibit synthesis by the isolated Pol ε holoenzyme (Schauer and O'Donnell, 2017). Consistent with this model, reducing the concentration of all 4 dNTPs to 5 μM for the duration of the reaction resulted in an RFC-dependent increase in DNA synthesis rate (Fig. EV3A). To test the model via another means, we performed a staged CMG-based assay where Pol ε was added 1 min after DNA unwinding was initiated (Fig. 3B). In this setting, RFC should be able to access the 3′ end of the leading-strand primer to load PCNA before Pol ε is added. Figure 3C shows that the inclusion of either 5 nM or 20 nM RFC had no effect under standard assay conditions but accelerated

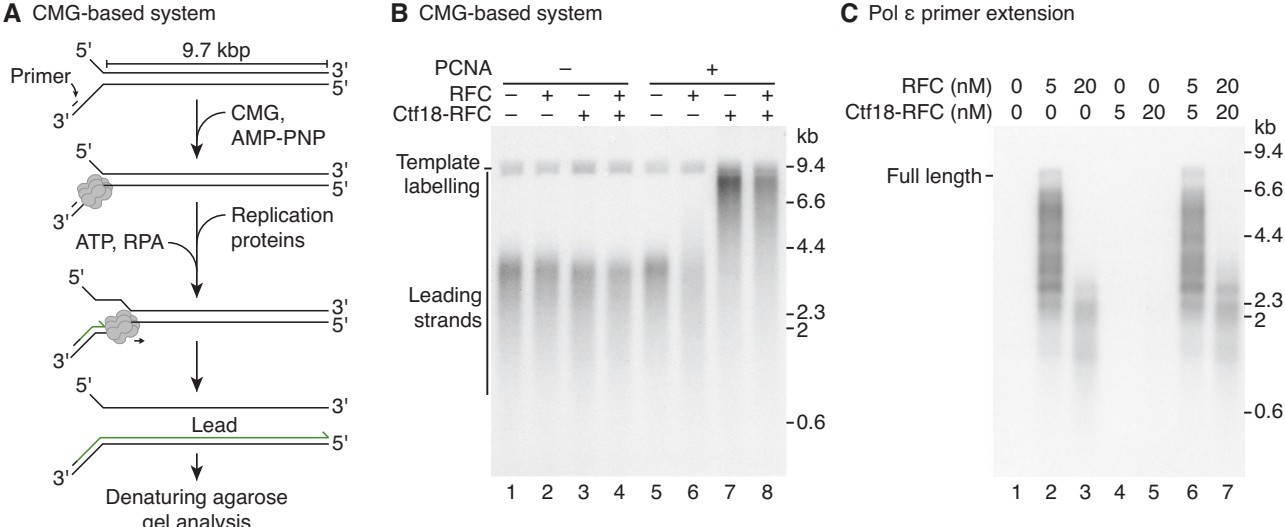

**Figure 2. Ctf18-RFC, but not RFC, supports PCNA loading at preassembled CMGE.**

(A) Schematic of the CMG-based in vitro DNA replication system. (B) Denaturing agarose gel analysis of replication reactions using the CMG-based system as in (A), in the absence or presence of PCNA, RFC, and Ctf18-RFC. Reactions were analysed after 3 min. (C) Denaturing agarose gel analysis of Pol ε primer extension reactions performed as in Fig. EV1B, in the presence of RFC or Ctf18-RFC at the concentrations indicated. Reactions were performed in the presence of 250 mM potassium glutamate and analysed after 6 min. Source data are available online for this figure.

leading-strand replication when the reaction was staged, and this acceleration was PCNA-dependent (Fig. EV3B). From these experiments, we conclude that when CMGE complexes are preassembled on primed replication forks, Pol ε binding to the 3′ end of the primer blocks access by RFC and this inhibition is maintained for the duration of replisome progression along the 9.7 kb template. In contrast to RFC, Ctf18-RFC can compete with Pol ε for access to the 3′ end to promote PCNA loading, either by capturing the 3′ end when Pol ε spontaneously dissociates, or potentially via a more coordinated hand-over mechanism.

## Pol ε functions with RFC-loaded PCNA after switch from Pol δ

In the regulated system, RFC efficiently loads PCNA for leading-strand synthesis by Pol ε (Fig. 1F), presumably because PCNA is either transferred from Pol δ to Pol ε during initiation, or by RFC gaining access to the 3′ end to load a new PCNA trimer after Pol δ has dissociated from the nascent leading strand, likely after colliding with the advancing CMGE complex (Fig. 4A) (Aria and Yeeles, 2018; Schauer and O'Donnell, 2017). To recapitulate this initiation mechanism in the CMG-based system we performed a staged reaction where Pol α-primase and Pol δ were added with Pol ε. Under these conditions, replisomes produced both leading- and lagging-strand products. Notably, the addition of Ctf18-RFC only had a modest effect on the reaction, increasing the maximal leading-strand synthesis rate from 3 to 3.5 kb min⁻¹ (Fig. 4B,C). This ~17% increase in rate is in line with the magnitude of Ctf18-RFC-dependent rate enhancement observed in the regulated system. Moreover, similar to the regulated system, in this set up RFC enhanced the abundance and rate of leading-strand products compared to a reaction lacking a clamp loader, whereas Ctf18-RFC only modulated the synthesis rate (Fig. 4D).

To further validate that we had successfully recapitulated a Pol δ to Pol ε switch, we performed a staged reaction where either wild-type Pol δ or a catalytically inactive complex (Pol δ^CAT-DEAD) were added, the rationale being that if Pol δ was participating in the establishment of leading-strand synthesis, Pol δ^CAT-DEAD should block replication (Aria and Yeeles, 2018). Accordingly, the addition of Pol δ^CAT-DEAD almost completely inhibited leading-strand replication by Pol ε (Fig. EV3C), suggesting that Pol δ was accessing the 3′ end of the leading strand at most if not all replication forks. Thus, our results from the regulated and CMG-based systems show that, although RFC can load PCNA to support leading-strand synthesis, this likely occurs during the early stages of synthesis. Moreover, the finding that Ctf18-RFC accelerates the maximal rate of leading-strand replication in both systems, even after RFC has loaded PCNA for Pol ε, indicates that there is a requirement for PCNA loading on the leading strand during replisome progression and that Ctf18-RFC is more proficient than RFC at doing this. Based on our observation that Ctf18-RFC, but not RFC, can load PCNA onto the leading strand when CMGE is preassembled, we suggest that Pol ε periodically engages the 3′ end of the leading strand in the absence of PCNA, and when this occurs Ctf18-RFC is required to load PCNA to re-establish maximal replication rates.

## Structure of Ctf18-RFC in the replisome

We reasoned that the positioning of Ctf18-RFC in the replisome might underlie its ability to compete with Pol ε to load PCNA onto the leading strand. Although the well-characterised interaction between the Ctf18-1-8 module and Pol2cat tethers Ctf18-RFC to the replication fork (Liu et al, 2020; Stokes et al, 2020), the ATPase module comprised of Ctf18 and Rfc2-5 is predicted to be flexibly tethered to the Ctf18 C-terminus and therefore its position, which is likely to be critical for PCNA loading, is unclear. Pol2cat, which

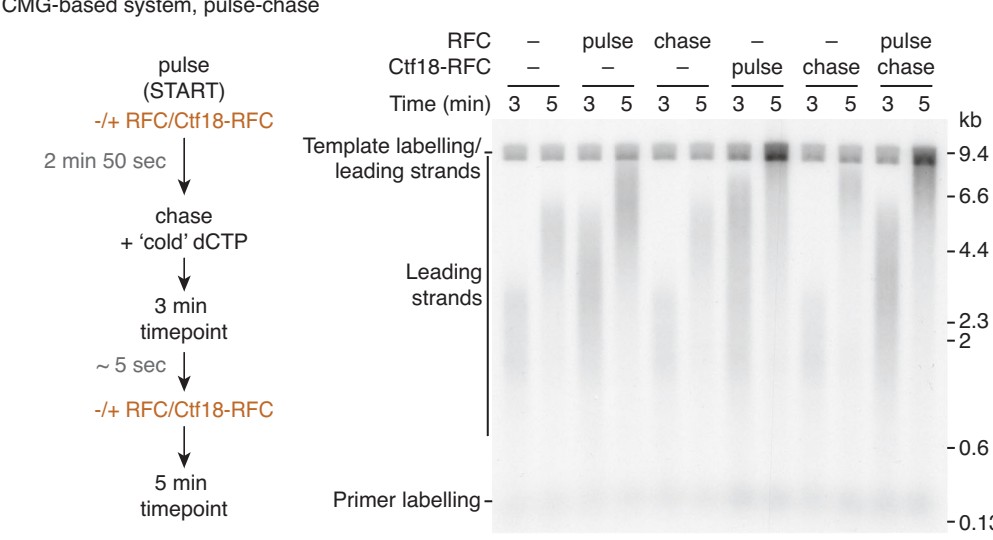

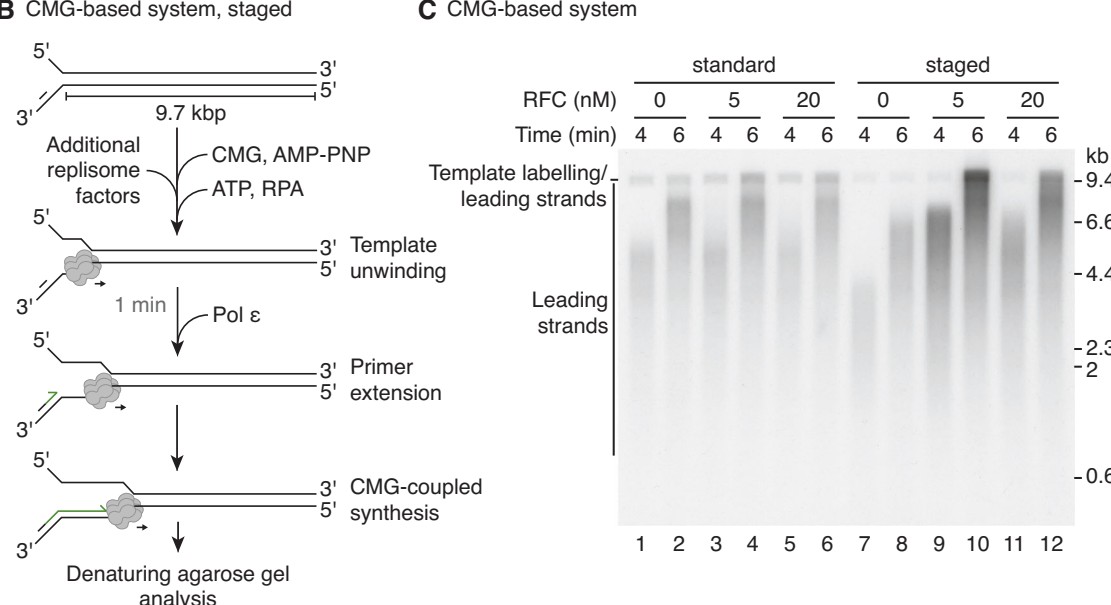

**Figure 3.  Transient uncoupling promotes PCNA loading by RFC.**

(A) Denaturing agarose gel analysis (right) of replication reactions using the CMG-based system under pulse-chase conditions as outlined (left). RFC and Ctf18-RFC were included during the pulse or after the chase where indicated. (B) Schematic of the staged CMG-based DNA replication system. Pol ε is added 1 min after initiation of template unwinding. (C) Denaturing agarose gel analysis of replication reactions using the standard CMG-based system as in Fig. 2A or the staged CMG-based system as in (B). RFC was included from the outset at the concentrations indicated. In this and subsequent staged CMG-based system reactions, timepoints are from initiation of template unwinding. Source data are available online for this figure.

consists of the first ~1200 amino acids of the Pol2 subunit (POLE1 in human), is flexibly tethered to the non-catalytic lobe of the holoenzyme complex (comprised of the C-terminal ~1000 amino acids of Pol2 together with Dpb2, Dpb3, and Dpb4) (Jones et al, 2021; Zhou et al, 2017). Consequently, Pol2$_{cat}$ displays considerable positional heterogeneity that makes obtaining 3D reconstructions from EM data challenging. Nevertheless, we previously observed Pol2$_{cat}$ and the human POLE1 catalytic domain (POLE1$_{cat}$) in two distinct positions when bound to CMG. In a human replisome

reconstruction, POLE1$_{cat}$ was rigidly associated with the non-catalytic domain such that the active site was situated ~140 Å from where the unwound leading-strand template emerges from the pore of CMG (Jones et al, 2021). In contrast, in a reconstruction of a budding yeast replisome with CMG encircling dsDNA to mimic an intermediate of replication termination, Pol2$_{cat}$ adopted a "folded" configuration positioning it directly under the Mcm2-7 hexameric ring so that the active site was only ~70 Å from where the leading-strand template emerges (Jenkyn-Bedford et al, 2021). How

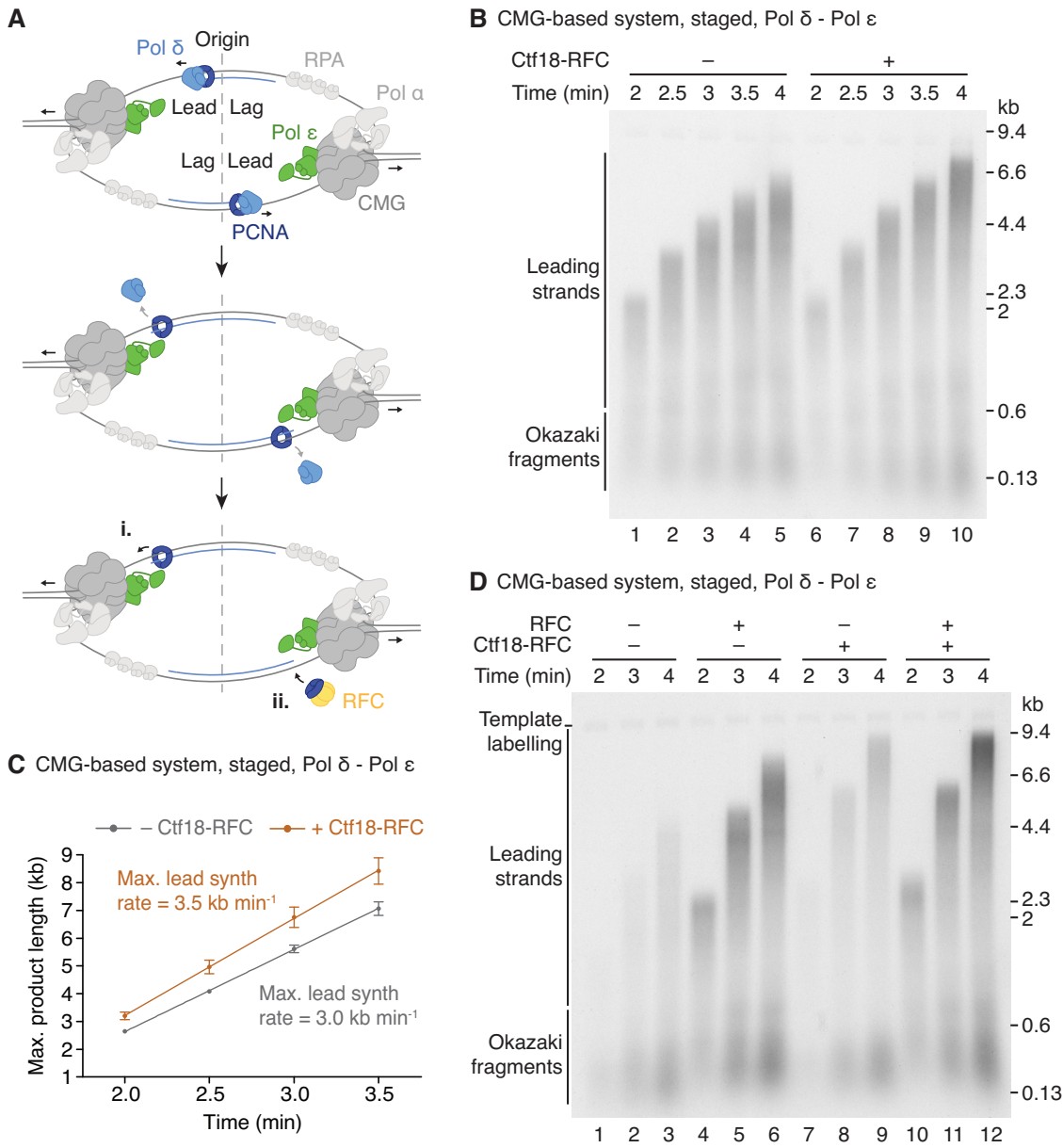

**Figure 4. Pol ε functions with RFC-loaded PCNA after switch from Pol δ.**

(A) Schematic representing polymerase exchange during replication initiation. Upon collision-release of extending Pol δ with CMGE, PCNA could either be transferred along with the 3′ DNA end (i) or loaded by RFC (ii). (B) Denaturing agarose gel analysis of replication reactions using the staged CMG-based system as in Fig. 3B, but with Pol ε, Pol δ, and Pol α-primase added 1 min after initiation of template unwinding. Ctf18-RFC was present where indicated, with RFC present throughout. (C) Quantification of maximal leading-strand synthesis rates from replication reactions in the staged CMG-based system as in (B). Linear regression is fit to the mean of three experiments. The error bars represent the s.e.m. and the mean is indicated by filled circles. (D) Denaturing agarose gel analysis of replication reactions using the staged CMG-based system with Pol ε, Pol δ and Pol α-primase added 1 min after initiation of template unwinding, in the absence or presence of RFC or Ctf18-RFC. Source data are available online for this figure.

these different configurations relate to Pol ε function is currently unclear.

To gain insights into the architecture and positioning of Ctf18-RFC when bound to Pol ε in the context of CMGE, we utilised a well-established pipeline (Baretic et al, 2020) to assemble a replisome complex containing CMG, Ctf4, Tof1-Csm3, Mrc1, Pol ε, Ctf18-RFC and forked DNA for analysis by cryo-EM (Fig. EV4). An initial 3D reconstruction using all particles that displayed good

cryo-EM density for CMG also displayed low-resolution signal "underneath" the C-tier of the MCM ring in a position reminiscent of the folded Pol2$_{cat}$ configuration we observed previously (Jenkyn-Bedford et al, 2021). Focused classification on the non-catalytic lobe of Pol ε and the region of density below MCM segregated ~30% of the particles into 3D classes with density displaying the characteristic features of Pol2$_{cat}$ (Appendix Figs. S1–S3). A similar approach using a focused mask based on the rigid conformation of

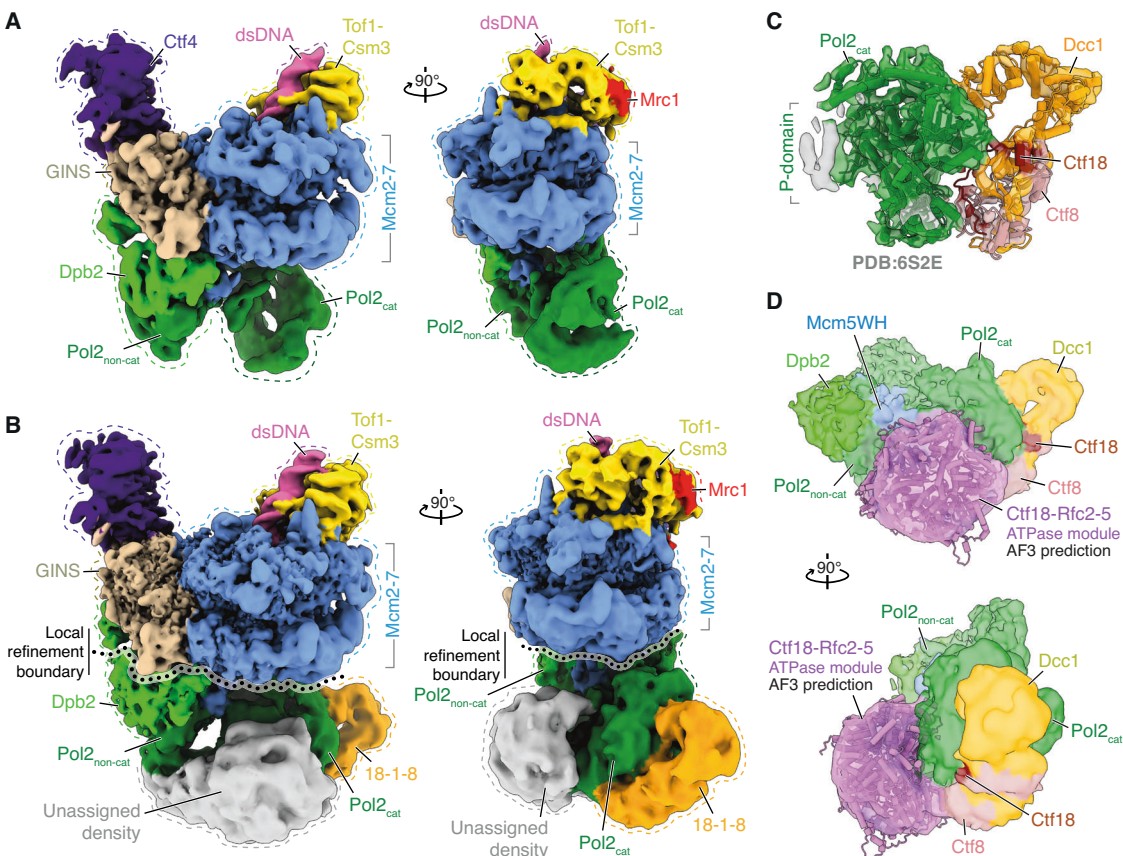

**Figure 5. Structural analysis of the budding yeast replisome with and without Ctf18-RFC.**

(A) Cryo-EM map of the replisome lacking density corresponding to Ctf18-RFC (EMD-52459). In this reconstruction, the Pol ε catalytic domain occupies a stable position beneath the Mcm2-7 C-tier. The map was generated via homogenous refinement and filtered according to local resolution. The map is coloured according to protein occupancy. (B) Cryo-EM maps of the replisome containing density corresponding to Ctf18-RFC. The individual maps displayed were obtained using local refinement following signal subtraction and filtered according to local resolution. Both maps are displayed on the same origin as the parental consensus refinement and describe the following replisome components respectively: CMG, Ctf4, DNA and Tof1-Csm3 (EMD-52107), and Pol ε and Ctf18-RFC (EMD-52116). The approximate boundary between these local refinements is indicated by a dotted line. Maps are coloured according to protein occupancy. (C) Cryo-EM map of the Pol ε catalytic domain bound by Ctf18, Dcc1, and Ctf8 (EMD-52120). This map was obtained via focused refinement following signal subtraction and sharpened using a B-factor of -300. PDB:6S2E (Stokes et al, 2020) was rigid body docked into the volume and the map coloured according to protein occupancy. (D) Cryo-EM map of Pol ε engaged with the Mcm5 WH domain and Ctf18-RFC (EMD-52116). This map was obtained via focused refinement following signal subtraction and filtered according to local resolution. The map was coloured according to protein occupancy. An AlphaFold 3 prediction for Ctf18-Rfc2-5 was manually docked into the cryo-EM density which remained unmodelled after the docking of Pol ε, the Mcm5 WH, and the Ctf18-1-8 module.

Pol ε failed to return any classes with density for Pol2cat. Further subclassification of the particles displaying Pol2cat density revealed two distinct classes of particles that were partitioned ~50:50 and that differed in the presence of an additional density alongside Pol2cat (Fig. 5A,B). This density was of sufficient resolution to unambiguously dock the Ctf18-1-8 module (Fig. 5C); the classes therefore represent replisomes with and without Ctf18-RFC bound to Pol2cat. Strikingly, when Ctf18-RFC was bound to Pol2cat via the Ctf18-1-8 module, reconstructions showed an additional large unmodelled density alongside Pol2cat and just below MCM of an appropriate shape and volume to accommodate an AlphaFold 3 model (Abramson et al, 2024) of the ATPase module of Ctf18-RFC (Fig. 5D). Because the ATPase and Ctf18-1-8 modules are predicted to be flexibly linked to each other, the ability to recover this additional density suggested that Ctf18-RFC might make additional protein:protein contacts within the replisome, specifically via its

ATPase module. In this regard, a cryo-EM structure of Pol ε–DNA–Ctf18-RFC–PCNA indicated that the Ctf18-RFC ATPase module can be stabilised through interactions with the Pol2cat N-terminal helix and P-domain (Yuan et al, 2024). However, docking of Pol2cat from this structure into our cryo-EM density does not position the Ctf18-RFC ATPase module in the unmodelled density and results in significant clashes with the MCM hexamer (Fig. EV5). Thus, the positioning of the Ctf18-RFC ATPase module in our replisome reconstruction is distinct from the Pol ε–DNA–Ctf18-RFC–PCNA structure.

## Interaction of Ctf18 with Mcm7 contributes to PCNA loading

Because the Ctf18-RFC ATPase module was positioned underneath the C-tier of MCM we hypothesised that one of the flexible

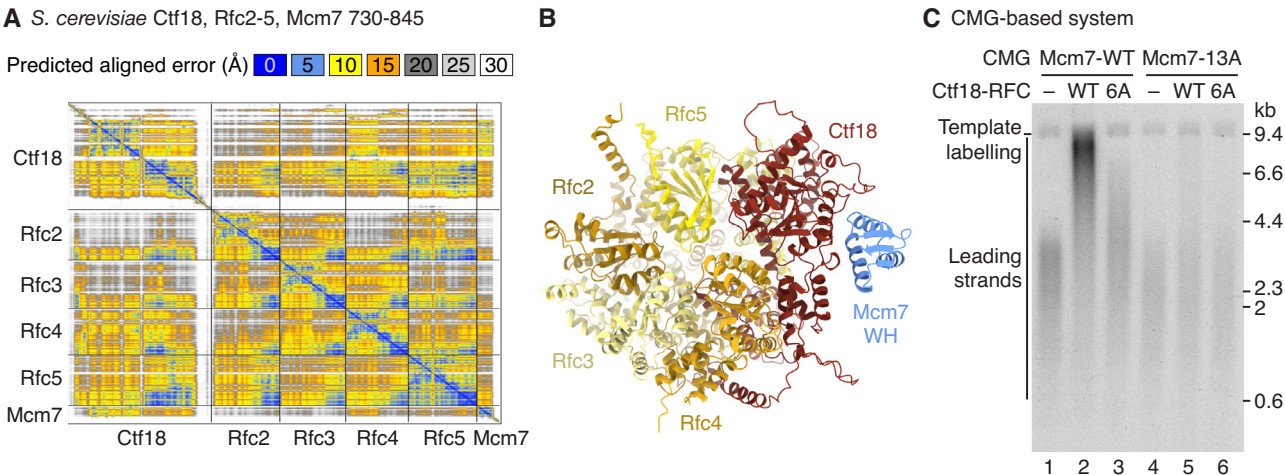

**Figure 6. Interaction of Ctf18 with Mcm7 is involved in PCNA loading.**

(A) AlphaFold 3 predicted aligned error plot of *S. cerevisiae* Ctf18, Rfc2-5 and Mcm7 730–845. (B) AlphaFold 3 prediction of *S. cerevisiae* Ctf18, Rfc2-5 and Mcm7 730–845 corresponding to (A), coloured by subunit. (C) Denaturing agarose gel analysis of replication reactions using the CMG-based system, with wild-type (WT) or mutant CMG and Ctf18-RFC complexes present where indicated. Reactions were analysed after 3 min. Source data are available online for this figure.

C-terminal winged-helix (WH) domains of the MCM subunits might interact with it. AlphaFold 3 structure predictions between the Ctf18-RFC ATPase module and each MCM WH domain (Appendix Figs. S4 and S5) identified a confident interaction between the Mcm7 WH and the Ctf18 ATPase domain (Figs. 6A,B and EV6A). A similar interaction was also predicted between human CTF18 and MCM7, but not between Mcm7 and the equivalent region of Rfc1, Elg1 or Rad24 (Appendix Fig. S5). To examine the significance of the putative interaction, we mutated residues that lie at the predicted interface; 6 amino acids in Ctf18 (Ctf18[6A]) and 13 in Mcm7 (Mcm7-13A) and purified Ctf18[6A]-RFC and CMG[Mcm7-13A], respectively (Fig. EV6B). Figure EV6C shows that Ctf18[6A]-RFC retained the ability to accelerate primer extension by Pol ε, demonstrating intact ATPase/ PCNA loading ability. CMG[Mcm7-13A] was proficient for DNA replication in the absence of a clamp loader and replication was stimulated by RFC in a staged CMG-based reaction (Fig. EV6D), demonstrating that replication with the Mcm7-13A mutant could be accelerated by leading-strand PCNA. We note that the rate of replication, both in the absence and presence of RFC, was slightly slower with CMG[Mcm7-13A], indicating that the Mcm7 WH might have a Ctf18-RFC-independent role during replisome progression. Crucially, in a standard CMG-based reaction, the ability of Ctf18-RFC to stimulate replication was significantly compromised by the Ctf18-6A mutations, and Ctf18-RFC had almost no effect on the reaction with CMG[Mcm7-13A] (Fig. 6C). These data strongly suggest that the predicted interaction between the Mcm7 WH and the Ctf18 ATPase domain is essential for leading-strand PCNA loading by Ctf18-RFC.

## Discussion

In vivo experiments have clearly demonstrated a requirement for PCNA loading by Ctf18-RFC on the leading strand in various replication-coupled processes (Bylund and Burgers, 2005;

Lengronne et al, 2006; Liu et al, 2020; Okimoto et al, 2016), and in vitro biochemistry and cryo-EM structures have revealed unique features of Ctf18-RFC that facilitate this process (Bermudez et al, 2003; Fujisawa et al, 2017; Grabarczyk et al, 2018; Murakami et al, 2010; Stokes et al, 2020; Yuan et al, 2024). Our work now shows that Ctf18-RFC can contribute directly to leading-strand synthesis in the budding yeast replisome, provides a mechanistic basis for why cells require Ctf18-RFC for leading-strand PCNA loading, and reveals additional mechanistic features that promote PCNA loading by Ctf18-RFC in the context of the replisome.

It has been reported that RFC and Pol ε display a similar affinity for primer-template DNA (Schauer and O'Donnell, 2017). Despite this, our data demonstrate that once CMG-bound Pol ε has engaged the 3′ end of the leading strand, RFC is almost completely blocked from productively loading PCNA for Pol ε, presumably because it is prevented from binding to the substrate by Pol ε. This behaviour is consistent with previous work that concluded that CMG protects Pol ε against RFC inhibition on the leading strand (Schauer and O'Donnell, 2017). In the absence of PCNA the isolated Pol ε holoenzyme has a processivity of ~20 nt (Hogg et al, 2014). Although it is plausible that interactions between Pol ε and replisome proteins might enhance its processivity, we consider it likely that the Pol ε catalytic domain frequently disengages from DNA during leading-strand replication but, due to it being tethered to the replisome, rapidly rebinds to continue synthesis. Notably, because slow DNA synthesis that was not accelerated by RFC/PCNA was observed along the entire length of the 9.7-kp template, Pol2[cat] might have transiently dissociated and rebound to the template hundreds of times without RFC gaining access to load PCNA. Thus, tethering of Pol ε to CMG prevents RFC from accessing the leading strand to load PCNA, which we propose underlies the requirement for a specialised leading-strand clamp loader (Ctf18-RFC) to ensure that the leading strand is populated with PCNA for replication-coupled processes.

Although we envisage that PCNA loading on the leading strand by RFC is negligible at established replication forks under optimal

conditions, our data indicate that RFC can gain access to the nascent leading strand to productively load PCNA when DNA synthesis is uncoupled from CMG-dependent template unwinding, which can be triggered by nucleotide depletion, DNA damage, and DNA secondary structure (Casas-Delucchi et al, 2022; Devbhandari and Remus, 2020; Kumar et al, 2021; Sparks et al, 2019; Taylor and Yeeles, 2018; Williams et al, 2023). Moreover, because leading-strand synthesis is initially uncoupled from CMG immediately after origin firing (Aria and Yeeles, 2018; Garbacz et al, 2018; Zhou et al, 2019), the first leading-strand PCNA will be loaded by RFC, which explains why RFC can support PCNA-dependent DNA replication in reconstituted origin-dependent budding yeast replication reactions (the regulated system).

How frequently does Ctf18-RFC load PCNA during leading-strand replication? On the one hand, our observation that Ctf18-RFC only had a modest effect on leading-strand synthesis rate suggests that PCNA loading might be relatively infrequent. However, PCNA loading on the leading strand would only be expected to influence replication rates at forks where CMGE had lost its association with PCNA. Yet, whether PCNA loss from CMGE is a prerequisite for Ctf18-RFC loading activity is unclear. Rather Ctf18-RFC might be presented with an opportunity to capture the 3′ end of the leading strand to load a new PCNA every time the 3′ end is vacated by Pol ε, even when Pol ε is functioning with PCNA. In this scenario, PCNA loading by Ctf18-RFC would not be expected to influence fork rate, provided that in its absence Pol ε can rebind to the 3′ end while maintaining or reestablishing its connection with PCNA. In the most simplistic version of this model, PCNA loading would be dictated by both the frequency of Pol ε dissociation from the 3′ end of the leading strand, which is currently unknown but could feasibly occur every ~500 nt based on the processivity of Pol ε in primer extension reactions with PCNA (Chilkova et al, 2007), and by the efficiency with which the 3′ end is captured by Ctf18-RFC. However, it has been proposed that the binding of the Ctf18-1-8 module to Pol ε weakens the association of the polymerase with DNA to facilitate DNA transfer to the Ctf18-RFC ATPase module (Yuan et al, 2024), and therefore Ctf18-RFC might not simply capture the 3′ end of the leading strand when Pol ε spontaneously lets go. Regardless of the precise reaction mechanism, we envision that single-molecule imaging of PCNA dynamics during leading-strand synthesis will be required to address how frequently PCNA is loaded.

To load PCNA the 3′ end of the nascent leading strand must be transferred from Pol ε to Ctf18-RFC. Currently, the position of Pol2$_{cat}$ during active leading-strand synthesis is unknown and, due to it being flexibly tethered to the remainder of Pol ε, has been visualised in multiple conformations in cryo-EM structures. Notably, our cryo-EM data shows Pol2$_{cat}$ positioned directly "underneath" the C-tier of the Mcm2-7 ring adjacent to where leading-strand template ssDNA exits the helicase. This configuration is observed independently of density corresponding to Ctf18-RFC and is reminiscent of a cryo-EM reconstruction of a terminated replisome that we previously observed (Jenkyn-Bedford et al, 2021). Although our replisome complexes were assembled in the absence of a nascent leading strand, we hypothesise that the location of Pol2$_{cat}$ underneath MCM approximates its position during active DNA replication because this arrangement minimises the length of ssDNA exposed between the helicase and polymerase by directing the leading strand template directly towards the polymerase active site.

Unexpectedly, our work has revealed that the interactions between Ctf18-RFC and Pol ε are not sufficient to support PCNA loading on the leading strand and that an additional replisome interaction, mediated by the C-terminal winged-helix of Mcm7 and the Ctf18 AAA + domain, is required. We propose that this interaction is needed to spatially constrain the Ctf18-RFC ATPase module to facilitate the capture of the 3′ end upon its dissociation from Pol2$_{cat}$. This idea is supported by the position of Pol2$_{cat}$ underneath MCM and the short unstructured linker predicted to connect the Mcm7 WH to its C-terminal ATPase domain. The interaction between the Mcm7 WH and Ctf18 appears to be conserved and was also identified as part of an Alphafold 2 protein:protein interaction screen of human proteins involved in genome maintenance (preprint: Schmid and Walter, 2024). This led the authors to propose that CTF18 positions the Pol ε catalytic domain so that leading-strand template DNA is fed directly into the polymerase. Although our data is consistent with this arrangement of Pol ε, it shows that tethering by CTF18 is not required for its adoption, at least in the yeast replisome, because we observed the Pol ε catalytic domain underneath MCM in 3D classes that lacked clear density for Ctf18-RFC, and in a prior study where Ctf18-RFC was absent (Jenkyn-Bedford et al, 2021). Assuming that the previously solved structure of Pol ε–DNA–Ctf18-RFC–PCNA (Yuan et al, 2024) represents an intermediate on the PCNA loading pathway, it is likely that the Pol ε catalytic domain must be repositioned to allow PCNA loading in the replisome because the structure is incompatible with the positioning of Pol2$_{cat}$ in our reconstruction.

Although considerable progress has been made towards understanding how Ctf18-RFC functions at replication forks to maintain genome stability, precisely how PCNA loading is coordinated in the context of the replisome remains an important unresolved question. Cryo-EM structures of PCNA loading in the context of the replisome will likely be required to address this and our work provides a strong platform for such investigations.

## Methods

**Reagents and tools table**

| Reagent/resource | Reference or source | Identifier or catalogue number |
|---|---|---|
| **Experimental models** | | |
| *E. coli* 5-alpha competent (high efficiency) | New England Biolabs | C2987H |
| yEF4 (Ctf18-RFC purification): *MATa ade2-1 ura3-1 his3-11,15 trp1-1 leu2-3,112 can1-100 bar1::Hyg pep4::KanMX ura3::URA3 pRS306/Rfc2-Gal-CBP-Tev-Rfc3 trp1::TRP1 pRS304/Rfc4-Gal-Rfc5 leu2::LEU2 pRS305/Ctf8-Gal-Dcc1 his3::HIS3 pRS303/Ctf18* | This study and (Williams et al, 2023) | N/A |

| Reagent/resource | Reference or source | Identifier or catalogue number |
|---|---|---|
| yEF10 (Ctf18$^{K189E}$-RFC purification): *MATa ade2-1 ura3-1 his3-11,15 trp1-1 leu2-3,112 can1-100 bar1::Hyg pep4::KanMX ura3::URA3 pRS306/Rfc2-Gal-CBP-Tev-Rfc3 trp1::TRP1 pRS304/Rfc4-Gal-Rfc5 leu2::LEU2 pRS305/Ctf8-Gal-Dcc1 his3::HIS3 pRS303/Ctf18$^{K189E}$* | This study | N/A |
| yEF16 (Ctf18$^{6A}$-RFC purification): *MATa ade2-1 ura3-1 his3-11,15 trp1-1 leu2-3,112 can1-100 bar1::Hyg pep4::KanMX ura3::URA3 pRS306/Rfc2-Gal-CBP-Tev-Rfc3 trp1::TRP1 pRS304/Rfc4-Gal-Rfc5 leu2::LEU2 pRS305/Ctf8-Gal-Dcc1 his3::HIS3 pRS303/Ctf18$^{K124A, L126A, E132A, R136A, R143A, Q199A}$* | This study | N/A |
| yEF18 (Ctf18-RFC$^{5A-R}$ purification): *MATa ade2-1 ura3-1 his3-11,15 trp1-1 leu2-3,112 can1-100 bar1::Hyg pep4::KanMX ura3::URA3 pRS306/Rfc2-Gal-CBP-Tev-Rfc3 trp1::TRP1 pRS304/Rfc4-Gal-Rfc5 his3::HIS3 pRS303/Ctf18$^{V730R, R731A, K732A}$ leu2::LEU2 pRS305/Ctf8-Gal-Dcc1$^{K364A, R367A, R380A}$* | This study | N/A |
| yEF26 (CMG$^{Mcm7-13A}$ purification): *MATa ade2-1 ura3-1 his3-11,15 trp1-1 leu2-3,112 can1-100 bar1::Hyg pep4::KanMX his3::HIS3 pRS303/Cdc45iFlag2 ade2::GINS-pat ura3::URA3 pRS306/Mcm2-Gal-CBP-TEV-Mcm3 trp1::TRP1 pRS304/Mcm4-Gal-Mcm5 leu2::LEU2 pRS305/Mcm6-Gal-Mcm7$^{E736A, D737A, S739A, T741A, T742A, K749A, E785A, Y786A, Y788A, N790A, D805A, D806A, D810A}$* | This study | N/A |
| yJF1 (Ctf18-RFC and CMG$^{Mcm7-13A}$ strain construction): *MATa ade2-1 ura3-1 his3-11,15 trp1-1 leu2-3,112 can1-100 bar1::Hyg pep4::KanMX* | Frigola et al, 2013 | N/A |
| yJY197 (wild-type CMG purification and Ctf4-depleted CMG strain construction): *MATa ade2-1 ura3-1 his3-11,15 trp1-1 leu2-3,112 can1-100 bar1::Hyg pep4::KanMX his3::HIS3 pRS303/Cdc45iFlag2 ade2::GINS-pat ura3::URA3 pRS306/Mcm2-Gal-CBP-TEV-Mcm3 trp1::TRP1 pRS304/Mcm4-Gal-Mcm5 leu2::LEU2 pRS305/Mcm6-Gal-Mcm7* | Jenkyn-Bedford et al, 2021 | N/A |
| yVA94 (Ctf4-depleted CMG purification): *MATa ade2-1 ura3-1 his3-11,15 trp1-1 leu2-3,112 can1-100 bar1::Hyg pep4::KanMX Ctf4-2xStrep-Nat his3::HIS3 pRS303/Cdc45iFlag2 ade2::GINS-pat ura3::URA3 pRS306/Mcm2-Gal-CBP-TEV-Mcm3 trp1::TRP1 pRS304/Mcm4-Gal-Mcm5 leu2::LEU2 pRS305/Mcm6-Gal-Mcm7* | This study | N/A |
| **Recombinant DNA** | | |
| pJF4: pRS305/Mcm6-Gal1-10-Mcm7 (mutagenesis of Mcm7 for CMG$^{Mcm7-13A}$ strain construction) | Frigola et al, 2013 | N/A |
| pRS304/Mcm4-Gal1-10-Mcm5 (CMG$^{Mcm7-13A}$ strain construction) | Frigola et al, 2013 | N/A |
| pRS304/Rfc4-Gal1-10-Rfc5 (Ctf18-RFC strain construction) | Yeeles et al, 2017 | N/A |
| pRS306/Mcm2-Gal1-10-Mcm3 (CMG$^{Mcm7-13A}$ strain construction) | Coster et al, 2014 | N/A |
| pRS306/Rfc2-Gal1-10-Rfc3 (Ctf18-RFC strain construction) | Yeeles et al, 2017 | N/A |
| vEF1: pRS303/Ctf18-Gal1-10-Gal4 (Ctf18-RFC strain construction) | This study | N/A |
| vEF17: pRS305/Ctf8-Gal1-10-Dcc1$^{K364A, R367A, R380A}$ (Ctf18-RFC$^{5A-R}$ strain construction) | This study | N/A |
| vEF18: pRS303/Ctf18$^{V730R, R731A, K732A}$-Gal1-10-Gal4 (Ctf18-RFC$^{5A-R}$ strain construction) | This study | N/A |
| vEF2: pRS305/Ctf8-Gal1-10-Dcc1 (Ctf18-RFC strain construction) | This study | N/A |
| vEF20: pRS303/Ctf18$^{K124A, L126A, E132A, R136A, R143A, Q199A}$-Gal1-10-Gal4 (Ctf18$^{6A}$-RFC strain construction) | This study | N/A |
| vEF26: pRS305/Mcm6-Gal1-10-Mcm7$^{E736A, D737A, S739A, T741A, T742A, K749A, E785A, Y786A, Y788A, N790A, D805A, D806A, D810A}$ (CMG$^{Mcm7-13A}$ strain construction) | This study | N/A |
| vEF4: pRS303/Ctf18$^{K189E}$-Gal1-10-Gal4 (Ctf18$^{K189E}$-RFC strain construction) | This study | N/A |
| vJY12: pRS303/Cdc45iFlag2 (CMG$^{Mcm7-13A}$ strain construction) | Jenkyn-Bedford et al, 2021 | N/A |
| vJY160: GINS-Pat (CMG$^{Mcm7-13A}$ strain construction) | Jenkyn-Bedford et al, 2021 | N/A |
| vVA23: pBP83 with Strep tag cloned in place of FLAG tag (tagging of endogenous Ctf4) | This study | N/A |
| vYE2: pMK-RQ/Dcc1 (mutagenesis of Dcc1 for Ctf18-RFC$^{5A-R}$ strain construction) | This study | N/A |
| **Oligonucleotides and other sequence-based reagents** | | |
| DBo1, lagging-strand oligonucleotide for structural reconstitution: GGCAGGCAGGCAGGCACACACTCTCCAATTCTCTAATCACTTACCA (BIOTIN-dT)CACTTCCTACTCTA | Baretic et al, 2020 | N/A |

| Reagent/resource | Reference or source | Identifier or catalogue number |
|---|---|---|
| DBo2, leading-strand oligonucleotide for structural reconstitution: Cy3-TAGAGTAGGAAGTGA(BIOTIN-dT)GGTAAGTGATTAGAGAATTGGA GAGTGTG(T)34 T∗T∗T∗T∗T ∗ denotes phosphorothioate backbone linkages | Baretic et al, 2020 | N/A |
| JY180, primer for M13mp18 ssDNA template: GAATAATGGAAGGG TTAGAACCTACCAT | Yeeles et al, 2017 | N/A |
| JY195, leading-strand primer for forked DNA template: CCTCTCGAGCCCATCCTTCCACTTCCCAACCCTCACC | Baris et al, 2022 | N/A |
| JY197, lagging-strand arm of forked DNA template: TTTTTTTTTTTTTTT TTTTTTCACACTCTCCAATTCTCACTTCCTACCACAT | Baris et al, 2022 | N/A |
| MT096, leading-strand arm of forked DNA template: /5Phos/ GCTATGTGGTAGGAAGTGAGAATTGGAGAGTGTGTTTTTTTTTT TTTTTTTTTTTTTTTTTTTTTTTTTTTTGAGGAAAGAATGTTGG TGAGGGTTGGGAAGTGGAAGGATGGGCTCGAGAGGTTTTTTTTTTTTT TTTTTTTTTTTTTTTTTTT | Baris et al, 2022 | N/A |
| M13mp18 ssDNA | New England Biolabs | N4040S |
| ZN3 | Taylor and Yeeles, 2018 | N/A |
| VA61, C-terminal tagging of endogenous Ctf4: TTAAAAAAATTAATAA TATAAGGGAA GCTAGATATGAACAGCAATTGAAACGTACGCTGCAGGTCGAC | This study | N/A |
| VA62, C-terminal tagging of endogenous Ctf4: TGAACAGGTATCAAAT AATTGTCTCTTGCGTATATATATTTTACATTTTTATCGATGAATT CGAGCTCG | This study | N/A |
| **Chemicals, enzymes, and other reagents** | | |
| [α-$^{32}$-P]-dCTP | Hartmann Analytic | SCP-205 |
| 3xFLAG peptide | Sigma | F4799 |
| Adenosine 5′-(b,g-imido)triphosphate (AMP-PNP) lithium salt hydrate | Sigma | A2647 |
| Adenosine 5′-triphosphate (ATP) (for protein purification) | Sigma | A7699 |
| Bovine serum albumin (BSA) | Invitrogen | AM2616 |
| COmplete, EDTA-free protease inhibitor tablets | Roche | 5056489001 |
| Dithiothreitol (DTT) | Melford | D11000 |
| dNTPs | Invitrogen | 10297018 |
| EDTA | Sigma | EDS |
| EGTA | VWR | 0732 |
| Glutaraldehyde | Sigma | G5882 |
| MOPS | Formedium | MOPS-SDS1000 |
| Nonidet P-40 substitute (NP-40-S) | Roche | 11754599001 |
| NTPs | ThermoFisher Scientific | R0481 |
| SeaKem LE agarose | Lonza | 50004 |
| Suberic acid bis(3-sulfo-N-hydroxysuccinimide ester) sodium salt (BS$^3$) | Sigma | S5799 |
| Tris(2-carboxyethyl) phosphine (TCEP) | Sigma | C4706 |
| TWEEN-20 | VWR | 663684B |
| Cdc45 | Yeeles et al, 2015 | N/A |
| Cdc6 | Coster et al, 2014 | N/A |
| Cdt1.Mcm2-7 | Coster et al, 2014 | N/A |
| CMG | Jenkyn-Bedford et al, 2021 | N/A |
| CMG$^{Mcm7-13A}$ | This study | N/A |
| Ctf18-RFC | This study | N/A |
| Ctf18-RFC$^{5A-R}$ | This study | N/A |
| Ctf18$^{6A}$-RFC | This study | N/A |
| Ctf18$^{K189E}$-RFC | This study | N/A |

| Reagent/resource | Reference or source | Identifier or catalogue number |
|---|---|---|
| Ctf4 | Yeeles et al, 2015 | N/A |
| DDK | On et al, 2014 | N/A |
| Dpb11 | Yeeles et al, 2015 | N/A |
| GINS | Yeeles et al, 2015 | N/A |
| Mcm10 | Yeeles et al, 2015 | N/A |
| Mrc1 | Yeeles et al, 2017 | N/A |
| ORC | Frigola et al, 2013 | N/A |
| PCNA | Yeeles et al, 2017 | N/A |
| Pol α-primase | Yeeles et al, 2017 | N/A |
| Pol δ | Yeeles et al, 2017 | N/A |
| Pol δ$^{CAT-DEAD}$ | Aria and Yeeles, 2018 | N/A |
| Pol ε | Yeeles et al, 2015 | N/A |
| Pol ε$^{exo-}$ | Guilliam and Yeeles, 2021 | N/A |
| Pol ε$^{PIP}$ | Aria and Yeeles, 2018 | N/A |
| RFC | Yeeles et al, 2017 | N/A |
| RPA | Baretic et al, 2020 | N/A |
| S-CDK (Δ1-100 Clb5) | Hill et al, 2020 | N/A |
| Sld2 | Yeeles et al, 2015 | N/A |
| Sld3/7 | Yeeles et al, 2015 | N/A |
| Tof1-Csm3 | Yeeles et al, 2017 | N/A |
| **Software** | | |
| AlphaFold 3 server | Abramson et al, 2024; Google DeepMind | https://alphafoldserver.com |
| Amersham Typhoon (1.1.0.7) | Cytiva | N/A |
| ChimeraX (v1.9) | UCSF Resource for Biocomputing, Visualisation and Informatics | https://www.cgl.ucsf.edu/chimerax/ |
| cryoSPARC (v3.0 - v4.2) | Structural Biotechnology | https://cryosparc.com/updates |
| Epson Scan 3.9.3.0EN | Seiko Epson Corporation | N/A |
| EPU (v2.0) | ThermoFisher Scientific (FEI) | https://www.fei.com/software/epu-automated-single-particles-software-for-life-sciences |
| ImageJ (v1.53k) | National Institute of Health | https://imagej.net/ij/ |
| Prism 9 | GraphPad | https://www.graphpad.com/updates/prism-900-release-notes |
| **Other** | | |
| 3.2 ml TLS-55 tube | Beranek Laborgerate | 362333 |
| Amersham Hyperfilm-MP | Cytiva | 28906844 |
| Anti-FLAG M2 Affinity Gel | Sigma | A2220 |
| BAS-IP MS phosphor screen | Cytiva | 28956474 |
| Calmodulin Sepharose 4B resin | Cytiva | 17052901 |
| Criterion XT 4–12% Bis-Tris precast gels | Biorad | 3450124 |
| Gradient-making station | Biocomp Instruments, Ltd | Part code: 108-2 |
| Illustra MicroSpin G-50 columns | Cytiva | 27533002 |
| MonoQ PC 1.6/5 | Cytiva | 17-0671-01 |
| MonoS 5/50 GL | Cytiva | 17-5168-01 |
| NuPAGE 4–12% Bis-Tris Mini Protein Gels | Invitrogen | NP0322 |

| Reagent/resource | Reference or source | Identifier or catalogue number |
|---|---|---|
| QUANTIFOIL Copper 400 mesh R2/2 holey carbon TEM grids | Electron Microscopy Sciences | Q450CR2 |
| Sephacryl S400 High-Resolution Gel | Cytiva | 17-0609-10 |
| SilverQuest Staining Kit | Invitrogen | LC6070 |
| Strep-Tactin XT Superflow high-capacity resin | IBA Lifesciences | 2-4030-025 |
| Superdex 200 Increase 10/300 GL | Cytiva | 28-9909-44 |
| Whatman 3 mm chromatography paper | Cytiva | 3030917 |

## Expression plasmid and yeast strain construction

Details of plasmids and vectors made for protein purification during this study can be found in the Reagents and Tools Table. Sequences for expression of Ctf18, Dcc1 and Ctf8 were codon optimised for overexpression in *S. cerevisiae*, synthesised by GeneArt synthesis, and subcloned into pRS expression vectors to produce vEF1 and vEF2. Ctf18[K189E] was obtained by site-directed mutagenesis of plasmid vEF1 to produce vEF4. Dcc1[3A] (K364A/R367A/R380A) was obtained by site-directed mutagenesis of plasmid vYE2 and subsequently subcloned into vEF2 to produce vEF17. Ctf18[RAA] (V730R/R731A/K732A) was obtained by site-directed mutagenesis of plasmid vEF1 to produce vEF18. Ctf18[6A] (K124A/L126A/E132A/R136A/R143A/Q199A) was obtained by subcloning a GeneArt synthesised fragment containing the relevant point mutations into vEF1 to produce vEF20. Mcm7[13A] (E736A/D737A/S739A/T741A/T742A/K749A/E785A/Y786A/Y788A/N790A/D805A/D806A/D810A) was obtained by subcloning a GeneArt synthesised fragment containing the relevant point mutations into pJF4 (pRS305/Mcm6-7) (Frigola et al, 2013) to produce vEF26. Sequences of plasmids made via subcloning/site-directed mutagenesis were verified via Sanger sequencing.

Details of yeast expression strains made during this study can be found in the Reagents and Tools Table. Wild-type and mutant Ctf18-RFC expression strains were produced by sequential transformation of yJF1 (Frigola et al, 2013) with pRS306/Rfc2-3 and pRS304/Rfc4-5 (Yeeles et al, 2017) followed by vEF2 (or vEF17) and vEF1 (or vEF4/vEF18/vEF20). For depletion of endogenous Ctf4 during CMG purification, a C-terminal 2xStrep tag was added to Ctf4 via transformation of yJY197 (Jenkyn-Bedford et al, 2021) with a PCR product generated from vVA23 (a modified pBP83 (Frigola et al, 2013) construct) using oligonucleotides VA61 and VA62. The CMG[Mcm7-13A] expression strain was produced by sequential transformation of yJF1 with vJY12 (pRS303/Cdc45iFlag2) (Jenkyn-Bedford et al, 2021), vJY160 (GINS-Pat) (Jenkyn-Bedford et al, 2021), pRS306/Mcm2-3 (Coster et al, 2014), pRS304/Mcm4-5 (Frigola et al, 2013) and vEF26.

## Replication protein expression and purification

An overview of protein purification strategies can be found in Appendix Table S1. Ctf18-RFC was purified from *S. cerevisiae* strain yEF4 as described previously for purification of RFC (Yeeles et al, 2017), apart from application to the Superdex 200 Increase 10/300 GL column which was performed in the same buffer but with the salt concentration increased to 300 mM NaCl. Ctf18[K189E]-RFC, Ctf18-RFC[5A-R], and Ctf18[6A]-RFC were expressed and purified as for wildtype from *S. cerevisiae* strains yEF10, yEF18 and yEF16, respectively.

CMG was purified as described previously (Baretic et al, 2020) but using *S. cerevisiae* haploid strain yJY197 (Jenkyn-Bedford et al, 2021). CMG[Mcm7-13A] was expressed and purified from yEF26 as for wild-type but with a twofold increase in protease inhibitors and all steps performed at 4 °C. Ctf4-depleted CMG for cryo-EM analysis was expressed and purified from yVA94 as for wildtype but to deplete endogenous Ctf4-2xStrep the FLAG elution was incubated with 0.6 ml Strep-Tactin XT Superflow resin for 30 min at 4 °C, the resin collected and flowthrough reapplied, and then flowthrough collected and used for subsequent purification steps.

## Primer extension reactions

To prepare primed ssDNA, reactions containing 10 mM Tris-HCl pH 7.5, 100 mM NaCl, 5 mM EDTA, 50 nM M13mp18 and 500 nM oligonucleotide JY180 (Reagents and Tools Table) were cooled gradually from 75 °C to room temperature. The unannealed oligonucleotide was removed using Sephacryl S400 High-Resolution Gel equilibrated in TE.

For primer extension, 1 nM DNA template was incubated with 500 nM RPA or where stated 400 nM *E. coli* SSB for 10 min at 30 °C in a reaction containing 25 mM HEPES-KOH pH 7.6, 10 mM Mg(OAc)$_2$, 100 mM potassium glutamate, 0.1 mg/ml BSA, 80 µM dA/dGTP, and 500 µM ATP. 20 nM PCNA, 20 nM RFC/Ctf18-RFC, and 20 nM Pol ε/10 nM Pol δ were added, and incubation was extended for 5 min. Reactions were initiated with the addition of 80 µM dT/dCTP and 33 nM [α-$^{32}$-P]-dCTP (final concentrations) and incubated at 30 °C. Aliquots were quenched with a final concentration of 25 mM EDTA at the specified timepoints.

## Regulated system replication reactions

Regulated system reactions were conducted as previously described using Ahd1-linearised CsCl gradient purified ZN3 plasmid (Taylor and Yeeles, 2018). MCM loading and phosphorylation were performed at 24 °C for 10 min in a reaction containing 25 mM HEPES-KOH pH 7.6, 100 mM potassium glutamate, 0.01% NP-40-S, 1 mM DTT, 10 mM Mg(OAc)$_2$, 0.1 mg/ml BSA, 40 mM KCl, 3 mM ATP, 3 nM Ahd1-linearised ZN3 DNA, 75 nM Cdt1-Mcm2-7, 40 nM Cdc6, 20 nM ORC, and 25 nM DDK. S-CDK was added to 80 nM, and incubation was extended for 5 min. The loading reaction was then diluted fourfold into replication buffer containing the following components (reported in their final concentrations): 25 mM HEPES-KOH pH 7.6, 250 mM potassium glutamate, 0.01% NP-40-S, 1 mM DTT, 10 mM Mg(OAc)$_2$, 0.1 mg/ml BSA, 3 mM ATP, 200 µM C/G/UTP, 30 µM dA/dG/dC/dTTP, and 33 nM [α-$^{32}$-P]-dCTP and incubation temperature increased to 30 °C. Reactions were initiated by addition of replication proteins to the

following concentrations: 30 nM Dpb11, 100 nM GINS, 30 nM Cdc45, 10 nM Mcm10, 20 nM Ctf4, 20 nM Tof1-Csm3, 20 nM PCNA, 20 nM Pol ε, 10 nM Pol δ, 100 nM RPA, 20 nM Pol α-primase, 10 nM Mrc1, 12.5 nM Sld3/7, 20 nM Sld2, 20 nM RFC, 20 nM Ctf18-RFC. Additional salt contributions from protein storage buffers ranged from approximately 35-45 mM. Aliquots were quenched with a final concentration of 50 mM EDTA at the specified timepoints.

## CMG-based system replication reactions

Forked linear DNA template with a leading-strand primer was prepared using CsCl gradient purified ZN3 plasmid as described previously (Baris et al, 2022). Briefly, plasmid ZN3 was SapI-linearised before ligation to a fork construct of oligonucleotides MT096, JY195 and JY197 (Reagents and Tools Table) with subsequent removal of excess unligated fork via Sepharose 4B gel filtration.

For CMG-based system reactions, CMG was loaded onto the DNA template at 30°C for 10 min in a reaction containing 25 mM HEPES-KOH pH 7.6, 100 mM potassium glutamate, 0.01% NP-40-S, 1 mM DTT, 10 mM Mg(OAc)$_2$, 0.1 mg/ml BSA, 2 nM primed forked DNA template, 100 µM AMP-PNP, and 50 nM CMG. The reaction was then diluted twofold into replication buffer containing the following components (reported in their final concentrations): 25 mM HEPES-KOH pH 7.6, 250 mM potassium glutamate, 0.01% NP-40-S, 1 mM DTT, 10 mM Mg(OAc)$_2$, 0.1 mg/ml BSA, 30 µM dA/dCTP, 20 nM Pol ε, 20 nM PCNA, 25 nM Ctf4, 10 nM Mrc1, 20 nM Tof1-Csm3, and 20 nM RFC/ Ctf18-RFC. Incubation was extended for 5 min before reactions were initiated by the addition of 4 mM ATP, 200 µM C/G/UTP, 30 µM dG/dTTP, 33 nM [α-$^{32}$-P]-dCTP, and 100 nM RPA. Aliquots were quenched with a final concentration of 50 mM EDTA at the specified timepoints. For pulse-chase experiments, the concentration of dCTP was reduced to 3 µM in the pulse and increased to 600 µM in the chase.

Staged CMG-based system replication reactions were performed using standard CMG-based system reaction conditions but Pol ε addition was delayed until 1 min after reaction initiation. Where indicated 10 nM Pol δ/ δ$^{CAT-DEAD}$ and 10 nM Pol α-primase were included with Pol ε addition. In all reactions 20 nM RFC was present unless otherwise stated.

## Reaction processing and gel analysis

Unincorporated nucleotides were removed from DNA replication reactions using Illustra MicroSpin G-50 columns. Reactions were analysed using 0.6% alkaline agarose gels in 30 mM NaOH and 2 mM EDTA for 16 h at 24 V. Gels were fixed with 2× washes in 5% trichloroacetic acid at 4°C and subsequently neutralised with 1 M Tris-HCl (pH 8.0) before drying onto 3 mm chromatography paper. Dried gels were exposed on BAS-IP MS Storage Phosphor Screens and imaged using an Amersham Typhoon phosphorimager, or autoradiographed with Amersham Hyperfilm-MP.

## Cryo-EM sample preparation

To generate fork DNA, equal volumes of oligonucleotides DBo1 and DBo2 (Reagents and Tools Table) were annealed by gradual cooling from 75°C to room temperature. The oligonucleotide stock

solutions were made in 25 mM HEPES-NaOH pH 7.5, 150 mM NaOAc, 2 mM Mg(OAc)$_2$, and 0.5 mM TCEP.

For replisome reconstitution, a reaction containing 25 mM HEPES-NaOH pH 7.6, 150 mM NaOAc, 15 mM Mg(OAc)$_2$, 0.5 mM TCEP, 0.5 mM AMP-PNP, 150 nM CMG (Ctf4-depleted), with a 1.2-fold molar excess over CMG of Pol ε$^{exo-}$, Ctf18-RFC, Tof1-Csm3, and Ctf4, and a 1.5-fold molar excess over CMG of fork DNA and Mrc1, was set up in a final volume of 310 µl. CMG was first incubated with the fork DNA on ice for 30 min in an 85 µl reaction. The remaining proteins were then added, and the reaction volume was adjusted to 310 µl before further incubation on ice for 45 min. 212 µl of the reaction was applied to glycerol gradients in the presence of a crosslinker, with the remaining 98 µl applied to a glycerol gradient in the absence of a crosslinker. Glycerol gradients were prepared as previously described (Jones et al, 2023): Buffer A (40 mM HEPES-NaOH, pH 7.5, 150 mM NaOAc, 0.5 mM TCEP, 10% v/v glycerol, 0.5 mM AMP-PNP and 3 mM Mg(OAc)$_2$) was layered on top of an equal volume of Buffer B (Buffer A except for 30% v/v glycerol, and 0.16% glutaraldehyde and 2 mM bis(sulfosuccinimidyl)suberate (BS$^3$) for gradients with crosslinker) in a 2.2 mL TLS-55 tube (Beranek Laborgerate) and gradients made using a gradient-making station (Biocomp Instruments, Ltd.) before cooling on ice. The samples were separated by centrifugation at 200,000 × g, 4°C for 2 h and 100 µl fractions manually collected. Subsamples of each fraction were analysed on 4–12% Bis-Tris SDS-PAGE gels and silver-stained using the SilverQuest staining kit. Desired crosslinked fractions were pooled and buffer exchanged against 25 mM HEPES-NaOH pH 7.5, 150 mM NaOAc, 3 mM Mg(OAc)$_2$, 0.5 mM TCEP, 0.1 mM AMP-PNP, 0.005% TWEEN-20 during six rounds of ultrafiltration at 21,000 × g, 4°C, for 1 min/round. Samples were then concentrated to ~5 µl and immediately used for cryo-EM grid preparation on QUANTIFOIL R2/2 grids coated in ~2 nm of continuous carbon.

## Cryo-EM data collection

A total of 9494 raw movies were acquired using a 300 keV Titan Krios microscope (FEI) equipped with a K3 direct electron detector (Gatan) operated in electron counting mode using the EPU-automated acquisition software (ThermoFisher) with "Faster Acquisition" mode (AFIS) enabled. A slit width of 20 eV was used for the BioQuantum energy filter. Data were collected in super-resolution mode bin 2 at an effective pixel size of 0.94 Å/pixel over a defocus range of -1.5 to −3.5 µm. Movies were dose-fractionated into 39 fractions over a 4 s exposure, resulting in a total dose of 36.8901e$^-$/Å$^2$.

## Cryo-EM data processing

The data processing pipeline outlined here is schematised in Appendix Figs. S1–S3. In total, 9494 39-fraction movies were aligned and dose-weighted (0.9222525 e$^-$/Å$^2$/fraction, 5 ×5 patches, 150 Å$^2$ B-factor) using RELION-4's implementation of a MotionCor2-like programme. These micrographs were then imported into cryoSPARC-3, and all future operations were carried out using this software. CTF parameters were estimated using the "Patch CTF estimation" job and 36 poor-quality micrographs were excluded from future processing using the "curate-exposures" job.

Particles were picked using the "Blob-picker" job using a minimum particle diameter of 200 Å and a maximum of 450 Å. After running the "Inspect particle picks" job and applying filters Local power >186.000 and Local power <1090.000, 2,739,509 particles were extracted into a 120-pixel box with a pixel size of 3.996 Å.

Two successive rounds of heterogeneous refinement were carried out using a 3D reference volume from a previously published budding yeast replisome cryo-EM reconstruction (EMD-10227) and a "junk" reference that was derived from noise. This resulted in the identification of 3D reconstructions representing 1,158,572 particles that contained secondary structure features and resembled previous replisome reconstructions. Following homogeneous refinement, these particles were then classified using 3D classification without alignment into 10 discrete classes, employing hard classification and PCA as the initialisation mode —this classification regime is subsequently used throughout. Five classes, comprising 493,520 particles, were selected which contained density for CMG, Tof1-Csm3, dsDNA, Ctf4, and Pol ε simultaneously. These particles were refined to the Nyquist resolution of 8.9 Å and this reconstruction used to generate a soft mask covering CMG, Tof1-Csm3, dsDNA and Ctf4 using the UCSF Chimera "segment-maps" function. This mask was used to subtract the signal from particles prior to carrying out 3D classification without alignment on the remaining density. Fourteen classes were obtained and three selected as containing density for the Pol ε catalytic domain representing 365,567 particles. These particles were reverted to their non-subtracted counterparts and the homogeneous refinement, particle subtraction, and 3D classification without refinement routine repeated, this time specifying nine classes. Five of these classes contained additional density for the Pol ε catalytic domain and one class contained density for both the Pol ε catalytic domain and the Ctf18-1-8 module. At this stage, the pipeline splits to deal with reconstructions that contain Ctf18-RFC and those that do not separately.

The five classes containing density for the Pol ε catalytic domain but not Ctf18-RFC, consisting of 123,075 particles, were then refined to the Nyquist resolution of 8.9 Å. These particles were re-extracted using a pixel size of 0.93 Å in a 650-pixel box and refined using homogeneous refinement to 3.4 Å resolution. A soft mask covering CMG, Tof1-Csm3, and Ctf4 was generated as previously described and used to carry out signal subtraction on this subset. The resulting signal comprising Pol ε and the Mcm5 WH was refined within a mask using local refinement to a resolution of 4.9 Å. 3D classification without alignment was then carried out on the subtracted particles, and a single class was selected containing a secondary structure in the Pol ε catalytic domain. The 20,462 particles in this class were refined to their non-subtracted counterparts and re-refined using homogeneous refinement to 3.82 Å resolution—this map is deposited as EMD-52459. The density for CMG, Tof1-Csm3, dsDNA and Ctf4 was again subtracted and the Pol ε and Mcm5 WH density refined using local refinement to 4.25 Å resolution—this map is deposited as EMD-52505.

Returning to the point at which the processing pipeline diverged, the 151,627 particles that occupied the single class containing Ctf18-1-8 module density were reverted to their non-subtracted counterparts and refined to the Nyquist frequency of 8.9 Å resolution. The density for CMG, Tof1-Csm3, dsDNA and Ctf4 was subtracted as previously described, and the remaining signal was locally refined to the Nyquist resolution of 8.9 Å resolution. These particles were then classified using

3D classification without alignment, and two classes containing density for the Ctf18-1-8 module were selected for further processing. These particles were re-extracted using a pixel size of 0.93 Å in a 650-pixel box and refined using homogeneous refinement to 3.61 Å resolution. To increase the signal-to-noise of the particles, the particles were downsampled to either 1.374 Å/pixel or 2.015 Å/pixel. Density for Pol ε, the Mcm5 WH, and the Ctf18-1-8 module were then subtracted from the 1.374 Å/pixel subset and the remaining signal locally refined to 3.34 Å resolution—this map is deposited as EMD-52107. Density for CMG, Tof1-Csm3, dsDNA, and Ctf4 were then subtracted from the 2.015 Å/pixel subset and the remaining signal locally refined to 5.98 Å resolution—this map is deposited as EMD-52116. The same routine was applied to the 2.015 Å/pixel subset with the additional subtraction of the Pol ε non-catalytic module generating a reconstruction at 7.1 Å resolution—this map is deposited as EMD-52120.

## AlphaFold

AlphaFold 3 structure predictions were determined using Alpha-Fold3 Server with the seed option set to Auto.

## Quantification and statistical analysis

For the calculation of maximal leading-strand synthesis rates, lane profiles were generated in ImageJ, and the data points were transferred to GraphPad Prism 9. For each time point, a straight line was manually fitted to the 'front' of each leading-strand population peak and to the lane baseline. The intercept of these two lines was taken as the migration position for maximal length leading-strand products. These migration positions were converted to product lengths by interpolation from a standard curve derived from the molecular weight marker. This standard curve was generated by plotting the migration position of the molecular weight marker bands against the Log10 of their length and fitting it to a linear regression. All leading-strand synthesis rates were obtained from three independent repeats.

## Data availability

The cryo-EM density maps used for model building have been deposited in the Electron Microscopy Data Bank (EMDB) (https://www.ebi.ac.uk/pdbe/emdb) under the following accession numbers. Reconstructions derived from consensus refinements containing density for Ctf18-Dcc1-Ctf8 bound to the Pol-ε catalytic module: EMD-52107: Local refinement of CMG, Ctf4, Tof1-Csm3, and DNA. EMD-52116: Local refinement of Pol-ε and Ctf18-RFC. EMD-52120: Local refinement of the Pol-ε catalytic module bound to Ctf18-Dcc1-Ctf8. Reconstructions lacking density for Ctf18-Dcc1-Ctf8 bound to the Pol-ε catalytic module: EMD-52459: Consensus refinement of the replisome. EMD-52505: Local refinement of Pol-ε. This study does not report any original code. Any additional information required to reanalyse the data reported in this paper is available from the corresponding author upon request.

The source data of this paper are collected in the following database record: biostudies:S-SCDT-10_1038-S44318-025-00386-4.

# Peer review information

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

## Acknowledgements

The authors thank S Chen, G Sharov, G Cannone, A Yeates and B Ashan for smooth running of the MRC LMB EM facility; and J Grimmett, T Darling and I Clayson for the maintenance of scientific computing facilities. The authors thank the LMB media facility for the preparation of budding yeast media, and members of the Yeeles lab for discussions and feedback. This work was supported by the MRC, as part of UK Research and Innovation (MRC grant MC_UP_1201/12 to JTPY).

## Author contributions

**Emma E Fletcher**: Conceptualisation; Validation; Investigation; Visualisation; Methodology; Writing—original draft; Writing—review and editing. **Morgan L Jones**: Validation; Investigation; Visualisation; Methodology; Writing—review and editing. **Joseph T P Yeeles**: Conceptualisation; Supervision; Funding acquisition; Writing—original draft; Writing—review and editing.

Source data underlying figure panels in this paper may have individual authorship assigned. Where available, figure panel/source data authorship is listed in the following database record: biostudies:S-SCDT-10_1038-S44318-025-00386-4.

## Disclosure and competing interests statement

The authors declare no competing interests.

# Expanded View Figures

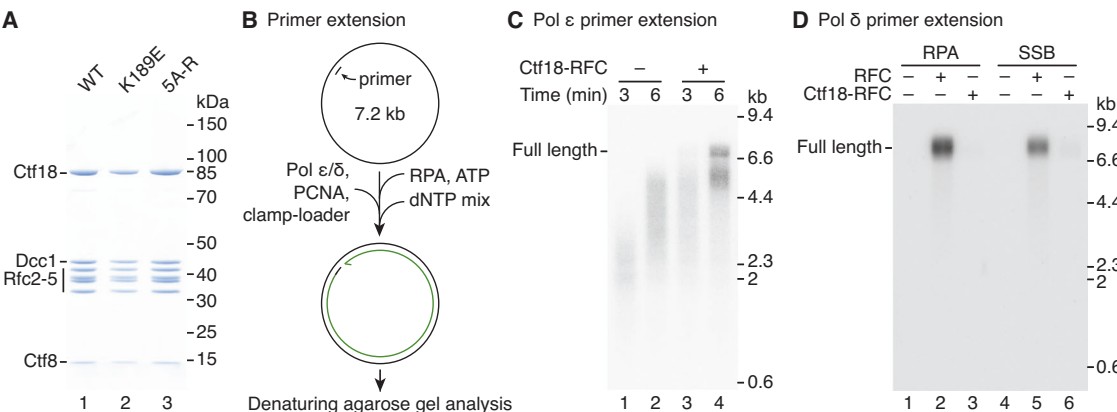

**Figure EV1. Ctf18-RFC mutants and analysis of Ctf18-RFC in primer extension assays.**

(A) Coomassie-stained 4–12% SDS-PAGE gel of purified *S. cerevisiae* wildtype (WT) or mutant Ctf18-RFC complexes. Subunits are labelled. (B) Schematic of a DNA polymerase primer extension reaction on circular ssDNA. (C) Denaturing agarose gel analysis of primer extension reactions as in (B) using Pol ε in the absence or presence of Ctf18-RFC. Reactions were performed at 100 mM potassium glutamate. (D) Denaturing agarose gel analysis of primer extension reactions as in (B) using Pol δ in the absence or presence of RFC or Ctf18-RFC, analysed after 1.5 min. The ssDNA template was coated with either *S. cerevisiae* RPA or *E. coli* SSB. Reactions were performed at 100 mM potassium glutamate. Source data are available online for this figure.

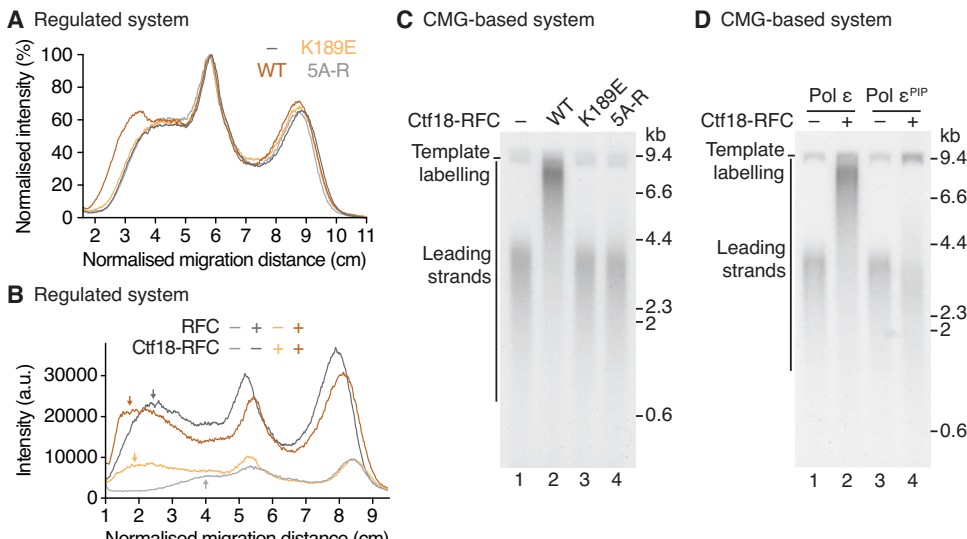

**Figure EV2.** **(Related to Figs. 1 and 2). Analysis of leading-strand synthesis acceleration by Ctf18-RFC in the regulated and CMG-based systems.**

(A) Lane profiles of 4 min timepoints from replication reactions using the regulated system, with wildtype (WT) or mutant Ctf18-RFC complexes present where indicated as in Fig. 1E. (B) Lane profiles of 5 min timepoints from replication reactions using the regulated system, in the absence and presence of RFC and Ctf18-RFC as in Fig. 1F. Arrows indicate 'left' leading-strand populations. (C) Denaturing agarose gel analysis of replication reactions using the CMG-based system, with wildtype (WT) or mutant Ctf18-RFC complexes present where indicated. Reactions were analysed after 3 min. (D) Denaturing agarose gel analysis of replication reactions using the CMG-based system with Pol ε or Pol ε$^{PIP}$, in the absence or presence of Ctf18-RFC. Reactions were analysed after 3 min. Source data are available online for this figure.

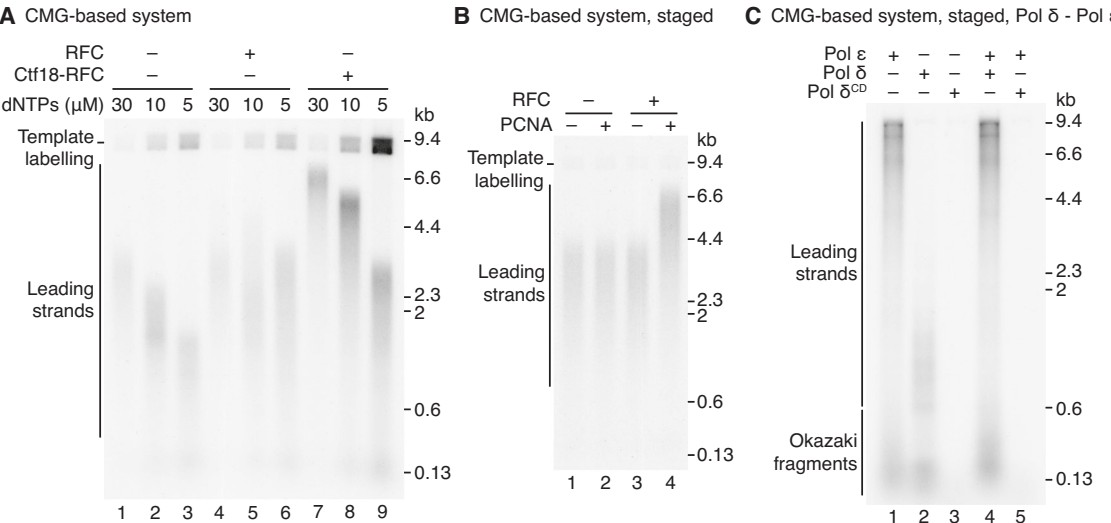

**Figure EV3.** **(Related to Figs. 3 and 4). Analysis of reduced dNTP and staged CMG-based system assays.**

(A) Denaturing agarose gel analysis of replication reactions using the CMG-based system in the absence or presence of RFC or Ctf18-RFC, with reduced dNTP concentrations as indicated. Reactions were analysed after 2.5 min. (B) Denaturing agarose gel analysis of replication reactions using the staged CMG-based system with Pol ε addition 1 min after initiation of template unwinding. PCNA and RFC were included where indicated. Reactions were analysed 4 min after initiation of template unwinding. (C) Denaturing agarose gel analysis of replication reactions using the staged CMG-based system but with Pol ε and Pol δ/ Pol δ$^{CAT\text{-}DEAD}$ (Pol δ$^{CD}$) added 1 min after initiation of template unwinding where indicated. Pol α-primase was included with polymerase addition throughout. Reactions were analysed 5 min after initiation of template unwinding. Source data are available online for this figure.

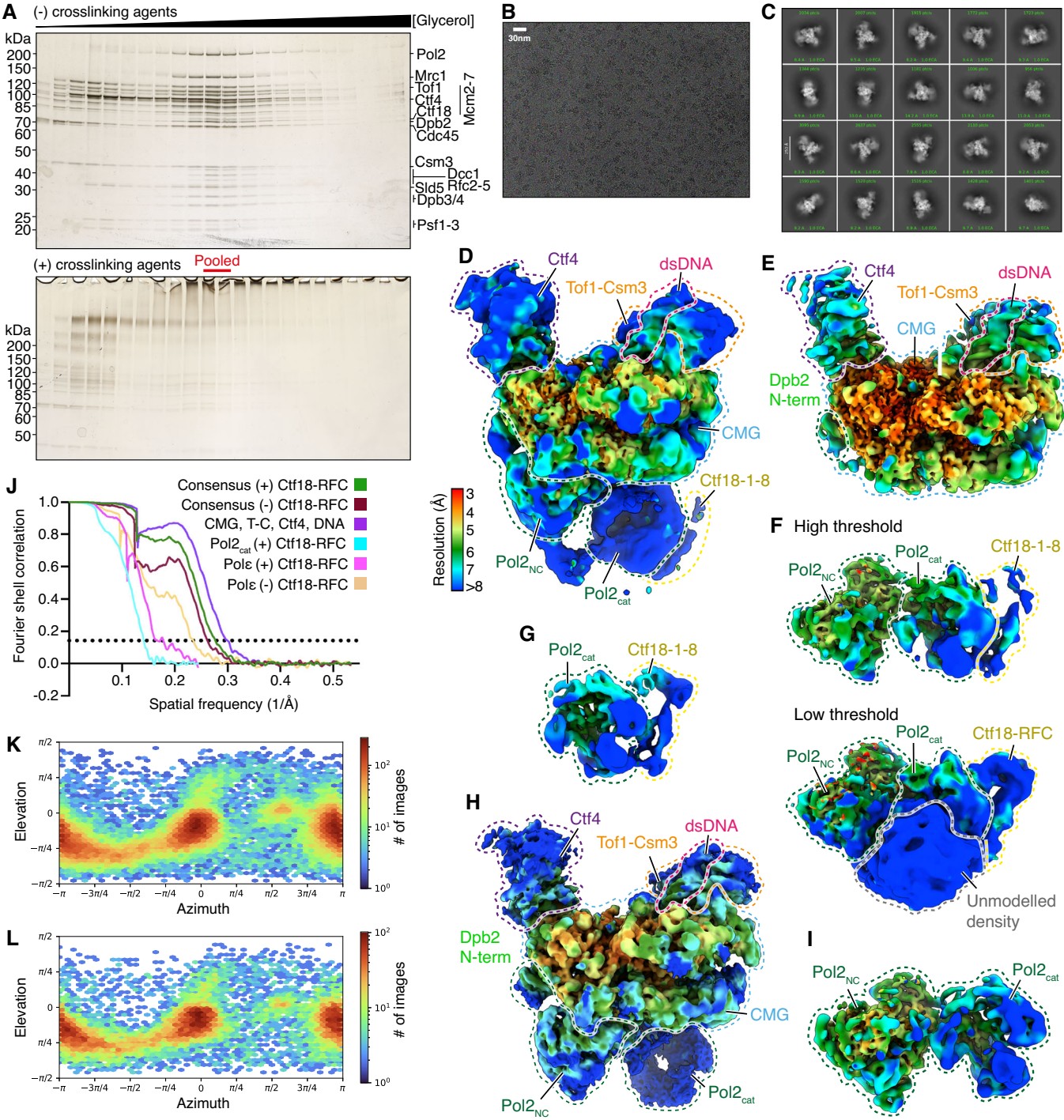

**Figure EV4.** (Related to Fig. 5). Cryo-EM analysis of a budding yeast replisome prepared with Ctf18-RFC.

(A) Silver-stained SDS-PAGE gels analysing 100 μl fractions taken across 10-30% glycerol gradients, either in the absence (top) or presence (bottom) of crosslinking agents. Fractions 11-12 used for cryo-EM sample preparation are indicated with a red bar labelled "pooled". Protein annotations are based on the position of bands in lane 12. (B) Representative cryo-EM micrograph obtained using a K3 direct electron detector (Gatan) at a nominal pixel size of 0.93 Å/pixel. Scale bar: 30 nm (inset). (C) Representative 2D class averages with corresponding particle numbers. Derived from 47,823 particle subset used to obtain EMD maps 52107, 52116, 52120. Mask diameter 500 Å. Obtained using cryoSPARC-3 2D classification. (D–I) Cryo-EM reconstructions obtained using homogeneous or local refinement, coloured according to local resolution according to the key in (D). Local resolution estimation and filtering was performed in cryoSPARC-3. (D) Replisome with Ctf18-RFC bound to the Pol ε catalytic domain. The 47,823 particle subset used for this homogenous refinement was also used to derive maps EMD-52107, EMD-52116 and EMD-52120. (E) Local refinement following particle subtraction showing CMG, Tof1-Csm3, dsDNA, Ctf4 and Dpb2$_{N-term}$. EMD-52107, derived from particle subset in (D). (F) Local refinement following particle subtraction, showing Pol ε, Ctf18-RFC and Mcm5 WH. EMD-52116, derived from particle subset in (D). (Top) map at high threshold, (bottom) same map at low threshold highlighting presence of unmodelled density. (G) Local refinement following particle subtraction, showing the Pol ε catalytic domain and Ctf18-RFC. EMD-52120, derived from particle subset in (D). (H) Replisome lacking Ctf18-RFC bound to the Pol ε catalytic domain. EMD-52459, obtained from a homogenous refinement of a 20,462 particle subset. (I) Local refinement following particle subtraction, showing Pol ε and Mcm5 WH. EMD-52505, derived from particle subset in (H). (J) Fourier shell correlation (FSC) graph describing the maps in (D–I). Resolution was calculated using the FSC = 0.143 cut-off with values reported in Appendix Figs. S1–S3. (K, L) Viewing direction plots. 2D-histograms that show the number of particles with a viewing direction at a particular elevation/azimuth bin. (K) Consensus refinement of the replisome with Ctf18-RFC bound, as in (D). (L) Consensus refinement of the replisome without Ctf18-RFC bound, as in (H). Source data are available online for this figure.

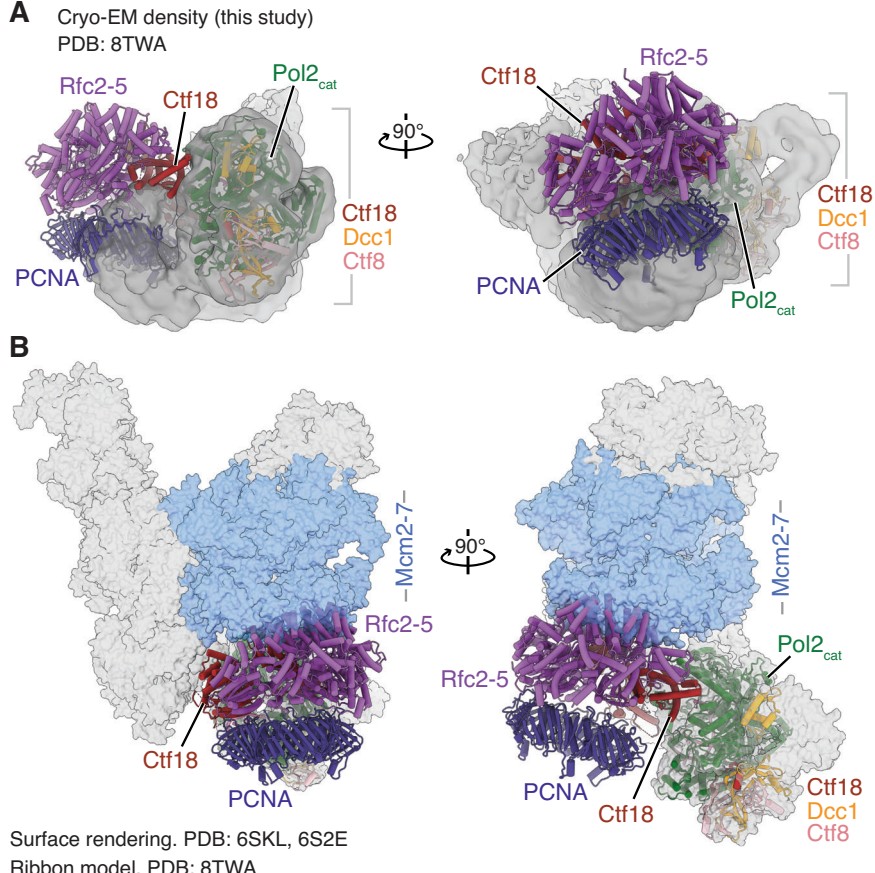

**Figure EV5.** (Related to Fig. 5). Structural comparisons of the replisome bound to Ctf18-RFC (this study) with relevant previously published structures.

(A) Model for Pol ε catalytic domain–DNA–Ctf18-RFC–PCNA complex (PDB:8TWA (Yuan et al, 2024)) docked into the cryo-EM density for Pol ε–Ctf18-RFC obtained in this study (EMD-52116). The Pol ε catalytic domain from 8TWA was rigid body docked into EMD-52116 using the "fit-in-map" command in ChimeraX. Docking highlights how the Ctf18-RFC ATPase module in 8TWA adopts an alternative position relative to Pol ε compared to that observed in this study. (B) Model for Pol ε catalytic domain–DNA–Ctf18-RFC–PCNA complex (PDB:8TWA) aligned to a model for the budding yeast replisome, where the Pol ε catalytic domain is positioned below the Mcm2-7 C-tier. To generate this replisome model, the previously published structure of the budding yeast replisome (PDB:6SKL (Baretic et al, 2020)) was rigid body docked into EMD-52107, and the Pol ε catalytic domain bound to Ctf18-RFC (PDB:6S2E (Stokes et al, 2020)) was docked into EMD-52116. The Pol ε catalytic domain of 8TWA was then aligned using Matchmaker in ChimeraX to the Pol ε catalytic domain of 6S2E. This alignment reveals how the conformation of the Ctf18-RFC ATPase module in 8TWA clashes with the C-tier of the Mcm2-7 helicase when the replisome adopts this configuration.

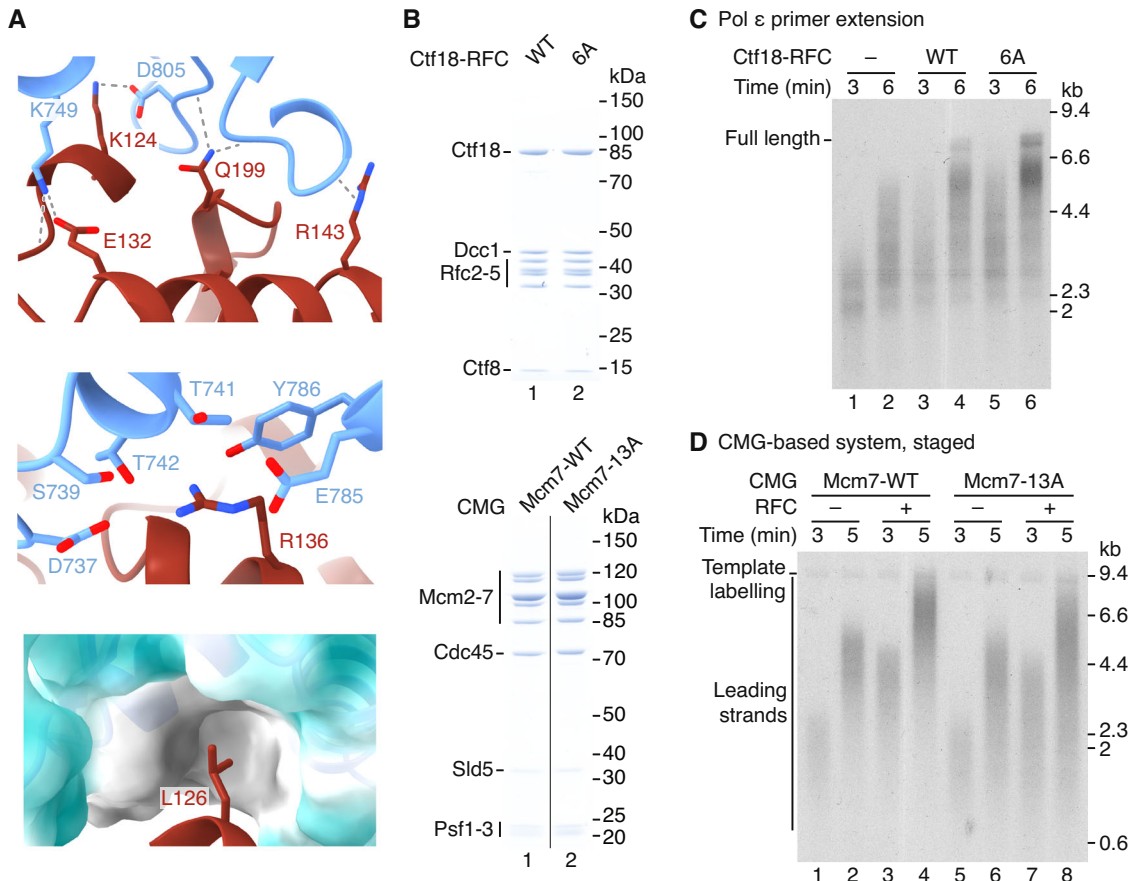

**Figure EV6.** **(Related to Fig. 6). Analysis of mutants designed to disrupt the interface between Ctf18 and the Mcm7 WH.**

(**A**) Detailed views of the AlphaFold predicted interaction between *S. cerevisiae* Ctf18 (red) and Mcm7 730–845 (blue). In the third panel, the Mcm7 WH molecular surface is rendered by hydrophobicity according to the Kyte-Doolittle scale, with hydrophilic regions in cyan and hydrophobic regions in grey. (**B**) Coomassie-stained 4–12% SDS-PAGE gel of purified *S. cerevisiae* wildtype (WT) or mutant Ctf18-RFC and CMG complexes. Subunits are labelled. (**C**) Denaturing agarose gel analysis of a Pol ε primer extension reaction with wildtype (WT) or mutant Ctf18-RFC complexes present where indicated. Reactions were performed at 100 mM potassium glutamate. (**D**) Denaturing agarose gel analysis of replication reactions using the staged CMG-based system with Pol ε addition 1 min after initiation of template unwinding. Wildtype (WT) or mutant CMG complexes and RFC were present where indicated. Source data are available online for this figure.

