## [Peer Review File · The EMBO Journal]

Competition for the nascent leading strand shapes the requirements for PCNA loading in the replisome

Emma Fletcher, Morgan Jones, and Joseph Yeeles

Corresponding author(s): Joseph Yeeles (jyeeles@mrc-lmb.cam.ac.uk)

Review Timeline:

Submission Date:	16th Oct 24
Editorial Decision:	17th Nov 24
Revision Received:	24th Jan 25
Accepted:	4th Feb 25

Editor: Hartmut Vodermaier

Transaction Report:

Dr. Joseph TP Yeeles
MRC Laboratory of Molecular Biology (LMB)
Francis Crick Avenue
Cambridge CB2 0QH
United Kingdom

17th Nov 2024

Re: EMBOJ-2024-119352
Competition for the nascent leading strand shapes the requirements for PCNA loading in the replisome

Dear Joe,

Thank you again for submitting your study on RFC and Ctf18 clamp-loaders in leading vs. lagging strand replication. I have now received the reports from three expert referees, copied below for your information, and I am happy to say that all of them found the study significant and overall convincingly conducted. Following addressing and incorporation of several specific comments and suggestions listed in the review, we would therefore like to consider a revised manuscript further for The EMBO Journal.

Please be reminded that it is our policy to allow only a single round of (major) revision, making it important to carefully respond to all points raised at this stage; therefore, do not hesitate to contact me already during the early stages of your revision work, in case you would like to discuss any of the issues raised by the reviewers, or if you should require an extension of the revision period.

Detailed information on preparing, formatting and uploading a revised manuscript can be found below and in our Guide to Authors. Thank you again for the opportunity to consider this work for The EMBO Journal, and I look forward to your revision in due time.

With kind regards,

Hartmut

9) To facilitate reproducibility and cross-laboratory adoption of methodologies, please structure the Materials & Methods section as outlined in our guide to authors, including a completed Reagents and Tools Table that can be downloaded from our author guidelines as well (<https://www.embopress.org/page/journal/14602075/authorguide#structuredmethods>).

10) Digital image enhancement is acceptable practice, as long as it accurately represents the original data and conforms to community standards. If a figure has been subjected to significant electronic manipulation, this must be clearly noted in the figure legend and/or the 'Materials and Methods' section. The editors reserve the right to request original versions of figures and the original images that were used to assemble the figure. Finally, we generally encourage uploading of numerical as well as gel/blot image source data; for details see: embopress.org/page/journal/14602075/authorguide#sourcedata

At EMBO Press, we ask authors to provide source data for the main manuscript figures. Our source data coordinator will contact you to discuss which figure panels we would need source data for and will also provide you with helpful tips on how to upload and organize the files.

In the interest of ensuring the conceptual advance provided by the work, we recommend submitting a revision within 3 months (15th Feb 2025). Please discuss the revision progress ahead of this time with the editor if you require more time to complete the revisions. Use the link below to submit your revision:

Link Not Available

Referee #1:

Description: This manuscript investigates the role of RFC and Ctf18 in loading PCNA into the replisome during leading and lagging strand replication. In my opinion, this is one of the more difficult questions to address in the replication field. The authors present evidence that "that once Pol ϵ is incorporated into the budding yeast replisome and has commenced leading-strand synthesis, RFC cannot load PCNA and this inhibitory state is maintained during replisome progression. By contrast, we find that Ctf18-RFC is uniquely equipped to load PCNA onto the leading strand, and show that this activity requires a direct interaction between Ctf18-RFC and the CMG (Cdc45-MCM-GINS) helicase."

Comments:

I believe the study deserves to be published. It investigates a very interesting distinction between the enzymology of leading and lagging strand replication of the eukaryotic nuclear genome, i.e., the roles of PCNA and its loading factors. The authors present substantial biochemical, cryo-EM and modeling data as evidence that Ctf18 specifically contributes to loading PCNA onto DNA for leading strand replication, potentially through an interaction with CMG. The data are clearly presented, they are novel, and they are consistent with the interpretations. They also lead to interesting speculations and to proposals for additional downstream studies.

A minor point for consideration:

Figure 1F - "and by 5 min leading strands were longer than in the reaction with RFC (Figure 1F, compare lanes 6 and 9)." The

stated length difference is difficult to see in the version I have. A longer exposure might help. Longer exposures for specific lanes here, and in other images, or at least regions of the lanes emphasized in the text, might help to reinforce the interpretations made in the text.

A thought for the future. In addition to obtaining a better understanding of the mechanisms by which PCNA is loaded by two distinct clamp loaders, I eventually look forward to the authors' ideas about the biological implications of the differences highlighted here between the loading of PCNA by the two clamp loaders.

Referee #2:

In the present work, Fletcher et al. use biochemical and structural approaches to clarify the mechanism by which budding yeast Ctf18 works with the Rfc2-5 complex to aid with PCNA loading during DNA replication. Ctf18-RFC is found to play a role in helping load PCNA for primer engagement by Pol epsilon (Pol-e) during leading strand synthesis that is distinct from canonical Rfc1-5 (RFC). Cryo-EM analysis indicates that Ctf18 (and its partner subunits Dcc1 and Ctf8) bind to Pol-e when the polymerase is associated with the replicative CMG helicase in a manner that would appear to allow the clamp-loading substituents of Ctf18-RFC to efficiently compete with Pol-e for primer/template binding.

Overall, the work is well organized and straightforward to follow. The data appear of good technical quality and support the stated conclusions; the findings in turn help clarify why eukaryotes have adopted specialized clamp loader subunit to specifically support leading strand replication and why the standard initiation and lagging-strand clamp loader complex is less efficient at this task. The work should broadly appeal to those in the replication field. Pending resolution of a few issues, publication in EMBO would seem warranted.

Primary comments:

Mcm7-Ctf18 interaction. The number of mutations introduced into both proteins are overly extensive (particularly the 13-residue change in Pol-e) to be confident that activity is not being lost for trivial reasons (e.g., local misfolding) that have nothing to do with a direct interaction. Is this contact seen in the EM maps? If so, please show. If not, some additional data are needed to buttress the claim for an interaction between the two regions. Alternatively, this section should be excised until the model can be more rigorously tested. As it stands, the findings are insufficiently developed to add to the rest of the work.

The cryoEM data processing workflow is quite state-of-the-art and comprehensive. This said, there are a few points that could use some clarification:

1. In Figure 5C, a focused refined map of Pol-e(cat) bound by Ctf18, Dcc1, and Ctf8 is shown, but the figure legend doesn't say which map it corresponds to in Figure EV5. (May be EMD-0002?) It would be helpful for all maps displayed in Figures 5 and EV4 to be assigned a unique number (like EMD-XXXX) and these numbers labeled both in Figures 5, EV4, and EV5.
2. Figure EV5 is a bit overwhelming. Can this be better organized? Also, it might be helpful to label which components remain in the reconstruction after subtraction and to color these components in the maps after 3D classification. Otherwise, it's somewhat hard to track the processing flowchart.
3. Fig. EV5, after the first step of 3D classification without alignment, only the first five classes were selected for further reconstruction because the other five classes didn't show density for CMG, Tof1-Csm3, dsDNA, CZ4, and Pol-e simultaneously. But perhaps in the discarded four classes (6/7/8/10), signal for 18-1-8 module and Pol-e(cat) may exist? Is it worth trying to use the discarded four classes and perform subtraction and focused refinement to see if there are states worth reporting?
4. The quality of maps (as least those shown) appear to have less featuredness than one might expect for the stated resolution, particularly map EMD-0001 and EMD-0002. Can these reconstructions be improved further or can additional views be presented that provide a better sense of the map quality?
5. Heterogeneous refinement was carried out using a 3D reference volume from a previously published budding yeast replisome cryo-EM reconstruction. This is a bit worrisome because CryoSPARC can suffer from reference bias when performing heterogeneous refinement even if the reference has been low pass filtered. It may be beneficial to run heterogeneous refinement using outputs from an ab-initio model as references, and to then compare the reconstructions to see whether the published reference has biased the refinement.

Line 207. If Pol-e prevents RFC from binding the 3' end throughout replisome progression as stated, then why should increasing RFC concentration lead to an increase in product formation (unless it's just by mass action)? The purpose of PCNA is ostensibly to aid in polymerase processivity - i.e., to keep it from falling off the 3' end so that synthesis can be maintained. However, if the polymerase is remaining bound to the 3' end throughout synthesis, then added RFC and PCNA should have no effect. Also, it's

not just product length but also abundance that is increasing. This would seem to imply that RFC can either rescue (activate) a set of stalled CMG-Pol-e's that fail to leave the primer or that RFC can somehow aid in recycling CMG-Pol-e's to forks that had escaped binding during the initial reaction setup. Please clarify.

Line 149. Comparison of lanes 6 and 9 suggest it is the abundance and not the rate that is affected when Ctf18 is present vs. RFC? The text states otherwise.

Lines 170-171. Contrary to the statement, it seems like there is a slight uptick in full length product formation when Pole-PIP is used with Ctf18?

Fig. EV3A. The claim that lower dNTP concentration leads to an increase in rate (as evidenced by the appearance of the full length leading strand product) is apparent from the gel. However, why does the distribution of smaller length products decrease at the same time?

Minor points:

Fig. 1E and 1F. It is difficult to see the difference between certain lanes (e.g., WT vs. mutant; + vs - Ctf18 when RFC is present). Please include plots as per Fig. 1C.

Fig. 1F. Why does the extent and level of leading strand synthesis look the same between + and - Ctf18 reactions?

Fig. 2. Why is Ctf18 needed for efficient synthesis in panel B but not panel C?

Fig. EV1C. Is RPA present in this reaction?

Fig. EV3A. What is template labeling and why is it increasing as dNTP concentration is lowered?

Line 317 and abstract. The effects of Ctf18 overall seem rather modest, at least in the budding yeast system, so it may not be wholly appropriate to say that cells 'require' Ctf18 for leading strand loading. Perhaps it would be better nuanced to say that the data help explain how Ctf18 can have a stimulatory effect on leading strand synthesis?

Referee #3:

This manuscript addresses the role of RFC and Ctf18-RFC at the eukaryotic replisome. Both clamp loaders are capable of loading PCNA, but the mechanistic basis for Ctf18-RFC's specific role in leading strand synthesis has remained unclear. A series of elegant biochemical experiments demonstrate that once Pol epsilon initiates leading strand synthesis, RFC is unable to load PCNA on the leading strand. In contrast, Ctf18-RFC successfully loads PCNA, a process that requires physical interactions between Ctf18, Pol epsilon, and CMG. Additionally, cryo-EM structures of the budding yeast replisome with Ctf18-RFC provide insights into the potential locations of the Pol epsilon catalytic domain and Ctf18-RFC. Interestingly, these structures, along with previously published works by Yuan et al. (2024) and Jenkyn-Bedford et al. (2021), suggest that despite a network of interactions, the Pol epsilon catalytic domain likely occupies different positions during PCNA loading and processive leading strand synthesis. I have no further comments and congratulate the authors on their work.

We are very grateful to the editors and referees for the considerable time spent reviewing and assessing our manuscript. We are very pleased with the positive responses to our work. Specific comments have been addressed in our revised manuscript and are detailed in our point-by-point responses below.

Referee #1:

Description: This manuscript investigates the role of RFC and Ctf18 in loading PCNA into the replisome during leading and lagging strand replication. In my opinion, this is one of the more difficult questions to address in the replication field. The authors present evidence that "that once Pol ϵ is incorporated into the budding yeast replisome and has commenced leading-strand synthesis, RFC cannot load PCNA and this inhibitory state is maintained during replisome progression. By contrast, we find that Ctf18-RFC is uniquely equipped to load PCNA onto the leading strand, and show that this activity requires a direct interaction between Ctf18-RFC and the CMG (Cdc45-MCM-GINS) helicase."

Comments:

I believe the study deserves to be published. It investigates a very interesting distinction between the enzymology of leading and lagging strand replication of the eukaryotic nuclear genome, i.e., the roles of PCNA and its loading factors. The authors present substantial biochemical, cryo-EM and modeling data as evidence that Ctf18 specifically contributes to loading PCNA onto DNA for leading strand replication, potentially through an interaction with CMG. The data are clearly presented, they are novel, and they are consistent with the interpretations. They also lead to interesting speculations and to proposals for additional downstream studies.

We are very pleased with the Referee's positive assessment of our manuscript.

A minor point for consideration:

Figure 1F - "and by 5 min leading strands were longer than in the reaction with RFC (Figure 1F, compare lanes 6 and 9)." The stated length difference is difficult to see in the version I have. A longer exposure might help. Longer exposures for specific lanes here, and in other images, or at least regions of the lanes emphasized in the text, might help to reinforce the interpretations made in the text.

We thank the Referee for raising this. It is indeed difficult to see a length difference in the products between lanes 6 and 9 in this experiment. We have therefore removed this statement. The effect of Ctf18-RFC on fork rate is more clearly illustrated by the quantitation of leading-strand synthesis rate in Figures 1B-1D.

A thought for the future. In addition to obtaining a better understanding of the mechanisms by which PCNA is loaded by two distinct clamp loaders, I eventually look forward to the authors' ideas about the biological implications of the differences highlighted here between the loading of PCNA by the two clamp loaders.

We agree that this is an interesting area for future studies.

Referee #2:

In the present work, Fletcher et al. use biochemical and structural approaches to clarify the mechanism by which budding yeast Ctf18 works with the Rfc2-5 complex to aid with PCNA loading during DNA replication. Ctf18-RFC is found to play a role in helping load PCNA for primer engagement by Pol epsilon (Pol- ϵ) during leading strand synthesis that is distinct from canonical Rfc1-5 (RFC). Cryo-EM analysis indicates that Ctf18 (and its partner subunits Dcc1 and Ctf8) bind to Pol- ϵ when the polymerase is associated with the replicative CMG helicase in a manner that would appear to allow the clamp-loading constituents of Ctf18-RFC to efficiently compete with Pol- ϵ for primer/template binding.

Overall, the work is well organized and straightforward to follow. The data appear of good technical quality and support the stated conclusions; the findings in turn help clarify why eukaryotes have adopted specialized clamp loader subunit to specifically support leading strand replication and why the standard initiation and lagging-strand clamp loader complex is less efficient at this task. The work should broadly appeal to those in the replication field. Pending resolution of a few issues, publication in EMBO would seem warranted.

We thank the referee for their positive assessment of our work and have attempted to address the points that they have raised in our revised manuscript.

Primary comments:

Mcm7-Ctf18 interaction. The number of mutations introduced into both proteins are overly extensive (particularly the 13-residue change in Pol-e) to be confident that activity is not being lost for trivial reasons (e.g., local misfolding) that have nothing to do with a direct interaction. Is this contact seen in the EM maps? If so, please show. If not, some additional data are needed to buttress the claim for an interaction between the two regions. Alternatively, this section should be excised until the model can be more rigorously tested. As it stands, the findings are insufficiently developed to add to the rest of the work.

The resolution of our EM maps was insufficient to identify the predicted contact between the Mcm7 WH and Ctf18. The Mcm7 WH is attached to the AAA+ domain via a flexible tether and we suspect that this is the primary reason that we were unable to recover higher resolution density for the Ctf18-RFC ATPase module bound to Mcm7 via its WH.

The reviewer rightly points out that we have made multiple amino acid substitutions to disrupt the predicted interaction between Ctf18 and Mcm7: 6 amino acids in Ctf18 and 13 amino acids in the Mcm7 winged helix. These substitutions were designed to maximize the possibility that the putative interaction would be disrupted. In this case we felt that making multiple amino acid substitutions was appropriate because we could perform control experiments that tested key activities of Ctf18-RFC and Mcm7 (in the context of CMG) to ensure that these activities were not affected by the substitutions. For the Ctf18-RFC mutant, we tested its ability to support PCNA-dependent DNA replication by Pol epsilon. This functionality requires that the Ctf18-RFC complex can bind to Pol epsilon and perform ATP-dependent PCNA loading onto a primer template. The Ctf18-RFC mutant displayed comparable activity to WT, indicating that the amino acid substitutions had little, if any, impact on Pol epsilon-dependent PCNA loading by Ctf18-RFC. Likewise, the CMG complex with the Mcm7 WH mutations supported PCNA-dependent DNA replication (with RFC as the clamp loader) at almost the same rate as the wild type complex. Therefore, the mutations clearly had minimal effect on the primary function of CMG (DNA helicase activity). Moreover, in the case of Mcm7, even if there was a degree of local protein unfolding of the winged helix domain, we do not think this would affect the interpretation of the data because we only saw significant defects in Ctf18-RFC-dependent aspects of replication.

We retain the belief that the manuscript is enhanced by the inclusion of the data regarding the putative Mcm7:Ctf18 interaction and its role in PCNA loading at the replisome. Although low resolution, the cryo-EM data is suggestive of additional interactions between Ctf18-RFC and the replisome and our functional data using mutants (based on AlphaFold predictions) indicates that this interaction is formed between Mcm7 and Ctf18 and that it contributes to PCNA loading at the fork. We think that this is a nice addition to the manuscript because it illustrates that the functioning of Ctf18-RFC at the replication fork is more complex than previously appreciated, and further highlights how Ctf18-RFC is specialised for leading-strand clamp loading, which is the main focus of the manuscript.

The cryoEM data processing workflow is quite state-of-the-art and comprehensive.

We are glad that the Referee appreciates our cryoEM data processing workflow - we did our best to include as much information as possible in the initial submission.

This said, there are a few points that could use some clarification:

1. In Figure 5C, a focused refined map of Pol-e(cat) bound by Ctf18, Dcc1, and Ctf8 is shown, but the figure legend doesn't say which map it corresponds to in Figure EV5. (May be EMD-0002?) It would be helpful for all maps displayed in Figures 5 and EV4 to be assigned a unique number (like EMD-XXXX) and these numbers labeled both in Figures 5, EV4, and EV5.

We appreciate the Referee's suggestion to enhance the clarity of the manuscript. To ensure that each cryo-EM map is clearly identified, we have updated the figure legends of Figure 5 and EV4 to include the unique EMD identifiers for all displayed maps. Furthermore, we have split EV5 into

three separate figures. Due to space constraints these figures are now included as supplementary data in the Appendix, including the relevant EMD identifiers for all deposited maps.

Regarding the specific query, the focused refined map of Pol- ϵ (cat) bound by Ctf18, Dcc1, and Ctf8 shown in Figure 5C corresponds to EMD-0003 in the original manuscript. This map has been reassigned as EMD-52120 in the revised manuscript.

2. Figure EV5 is a bit overwhelming. Can this be better organized? Also, it might be helpful to label which components remain in the reconstruction after subtraction and to color these components in the maps after 3D classification. Otherwise, it's somewhat hard to track the processing flowchart.

We agree with the reviewer that Figure EV5 is complex and could benefit from better organisation. As mentioned above, we have reorganised Figure EV5 into three separate supplementary figures:

- The first supplementary figure outlines the data processing workflow up to the point where the dataset is split based on the presence of density for Ctf18-Dcc1-Ctf8 bound to Pol- ϵ (cat).
- The subsequent two supplementary figures detail the data processing workflows for each resulting data subset individually.

Additionally, we have included coloured annotations to clearly indicate which protein components are present in selected 3D reconstructions to improve clarity.

3. Fig. EV5, after the first step of 3D classification without alignment, only the first five classes were selected for further reconstruction because the other five classes didn't show density for CMG, Tof1-Csm3, dsDNA, CZ4, and Pol-e simultaneously. But perhaps in the discarded four classes (6/7/8/10), signal for 18-1-8 module and Pol-e(cat) may exist? Is it worth trying to use the discarded four classes and perform subtraction and focused refinement to see if there are states worth reporting?

As part of a related project utilising the same cryo-EM dataset, we have done as the reviewer suggested and included all particles containing Pol- ϵ (non-cat) density, regardless of the occupancy of other replisome components, from the first step of 3D classification without alignment. However, this analysis did not reveal any additional states of Pol- ϵ or Ctf18-RFC. We observe that the resolution of the Ctf18-Rfc2-5 module is limited by continuous flexibility rather than particle number. Consequently, increasing the number of particles is very unlikely to enhance the resolution of this region or enable the identification of stable sub-classes. Furthermore, to ensure the relevance of our structural conclusions, we obtained reconstructions that include all factors present in the in vitro reconstituted biochemical reactions performed.

4. The quality of maps (as least those shown) appear to have less featuredness than one might expect for the stated resolution, particularly map EMD-0001 and EMD-0002. Can these reconstructions be improved further or can additional views be presented that provide a better sense of the map quality?

The reconstructions could not be significantly improved using currently available processing approaches. The lack of "featuredness" noted by the reviewer may refer to their interpretation of the EMD-deposited maps shown in Figure EV5. These images display the unsharpened reconstructions in which some of the high-resolution features associated with the reported resolutions are not identifiable. To better assess the quality of the maps and appreciate the visible features, we encourage the reviewer to download and inspect the maps directly. For this purpose, we have included half-maps, unsharpened, sharpened, and locally filtered reconstructions in the submission, and these files will be accessible to readers through the EMDB. All reported resolutions have been validated using the gold standard Fourier Shell Correlation (FSC=0.143) approach. Additionally, submission to the EMDB provides an independent validation of the resolution reported by the user. Validation reports describing this process were included in the original submission and will also be publicly available via the EMDB.

5. Heterogeneous refinement was carried out using a 3D reference volume from a previously published budding yeast replisome cryo-EM reconstruction. This is a bit worrisome because CryoSPARC can suffer from reference bias when performing heterogenous refinement even if the reference has been low pass filtered. It may be beneficial to run heterogeneous refinement using outputs from an ab-initio model as references, and to then compare the reconstructions to see whether the published reference has biased the refinement.

We have performed heterogeneous refinement with references generated externally (as described in the manuscript) and using ab-initio reconstructions from the same dataset. The results were identical.

Line 207. If Pol-e prevents RFC from binding the 3' end throughout replisome progression as stated, then why should increasing RFC concentration lead to an increase in product formation (unless it's just by mass action)? The purpose of PCNA is ostensibly to aid in polymerase processivity - i.e., to keep it from falling off the 3' end so that synthesis can be maintained. However, if the polymerase is remaining bound to the 3' end throughout synthesis, then added RFC and PCNA should have no effect. Also, it's not just product length but also abundance that is increasing. This would seem to imply that RFC can either rescue (activate) a set of stalled CMG-Pol-e's that fail to leave the primer or that RFC can somehow aid in recycling CMG-Pol-e's to forks that had escaped binding during the initial reaction setup. Please clarify.

We thank the reviewer for raising an important point. We did not intend to give the impression that we believe that Pol epsilon is bound tightly to the 3' end for the duration of the reaction. Rather, we think that it periodically dissociates from the 3' end but, because it is held in close proximity by virtue of its interactions with CMG, it rapidly rebinds and RFC cannot gain access. We have tried to clarify this in the text.

Line 149. Comparison of lanes 6 and 9 suggest it is the abundance and not the rate that is affected when Ctf18 is present vs. RFC? The text states otherwise.

We agree that the lanes in question cannot be used to make comparisons about rate and have removed the statement from the text.

Lines 170-171. Contrary to the statement, it seems like there is a slight uptick in full length product formation when Pole-PIP is used with Ctf18?

We thank the referee for raising this point. Our description of these data was not as accurate as it should have been. We have updated the text accordingly, emphasising that the extent to which Ctf18-RFC can stimulate replication by Pol e PIP is greatly diminished relative to wild type Pol epsilon.

Fig. EV3A. The claim that lower dNTP concentration leads to an increase in rate (as evidenced by the appearance of the full length leading strand product) is apparent from the gel. However, why does the distribution of smaller length products decrease at the same time?

We think that the referee might be confusing replisome-independent template labelling and full-length products. In this experiment no full-length products were synthesised in any lane. The smaller length products are the leading-strand products we are interested in. The key point is that lowering dNTP concentration from 30 to 5 μ M reduces the maximal rate in the absence of a clamp loader and with Ctf18-RFC, but not with RFC.

Minor points:

Fig. 1E and 1F. It is difficult to see the difference between certain lanes (e.g., WT vs. mutant; + vs - Ctf18 when RFC is present). Please include plots as per Fig. 1C.

We have included the requested plots in Figure EV2A and EV2B.

Fig. 1F. Why does the extent and level of leading strand synthesis look the same between + and - Ctf18 reactions?

In the regulated system, because RFC can load PCNA onto the leading strand during initiation, Ctf18-RFC only has a modest effect on replication by increasing the maximal rate of synthesis that we observe by ~15%. This increase in rate is most clearly visualized in Figs. 1B-D.

Fig. 2. Why is Ctf18 needed for efficient synthesis in panel B but not panel C?

Ctf18 is not needed for efficient synthesis in panel B, it is required to enhance the rate of synthesis. Panel B is a CMG-based DNA replication reaction. Here, Pol epsilon performs more rapid synthesis when functioning with PCNA and Ctf18-RFC, but not RFC, is able to load PCNA in this reaction set up. The experiment in panel C is a simpler primer extension assay. Here, both RFC and Ctf18-RFC can load PCNA for use by Pol epsilon.

Fig. EV1C. Is RPA present in this reaction?

RPA is present in all experiments unless stated otherwise.

Fig. EV3A. What is template labeling and why is it increasing as dNTP concentration is lowered?

Template labelling is replisome-independent radiolabeling of the linear template used in the DNA replication reactions. We cannot be sure of its precise mechanism but we suspect it is due to the activity of DNA polymerases at the ends of templates filling in overhangs, or perhaps also filling in overhangs that are generated by the DNA polymerase exonuclease activity. Template labelling likely increases at lower dNTP concentrations because lowering the total dNTP concentration results in an increase in specific activity of the radiolabel as the volume of radiolabel added to each sample was kept constant. This also results in an increase in the intensity of replisome-dependent products, but this is much more complicated due to their variable length, which also influences product intensity.

Line 317 and abstract. The effects of Ctf18 overall seem rather modest, at least in the budding yeast system, so it may not be wholly appropriate to say that cells 'require' Ctf18 for leading strand loading. Perhaps it would be better nuanced to say that the data help explain how Ctf18 can have a stimulatory effect on leading strand synthesis?

We fully agree that our data shows that Ctf18-RFC only makes a modest contribution to DNA replication rate in the in vitro yeast DNA replication systems. There is abundant evidence that Ctf18-RFC is crucial for multiple genome maintenance pathways in yeast and that these involve its PCNA loading activity. Our work helps to explain why this activity is required by establishing that RFC alone cannot fulfill this role because of the competition that it faces from Pol epsilon for access to the leading strand.

Referee #3:

This manuscript addresses the role of RFC and Ctf18-RFC at the eukaryotic replisome. Both clamp loaders are capable of loading PCNA, but the mechanistic basis for Ctf18-RFC's specific role in leading strand synthesis has remained unclear. A series of elegant biochemical experiments demonstrate that once Pol epsilon initiates leading strand synthesis, RFC is unable to load PCNA on the leading strand. In contrast, Ctf18-RFC successfully loads PCNA, a process that requires physical interactions between Ctf18, Pol epsilon, and CMG. Additionally, cryo-EM structures of the budding yeast replisome with Ctf18-RFC provide insights into the potential locations of the Pol epsilon catalytic domain and Ctf18-RFC. Interestingly, these structures, along with previously published works by Yuan et al. (2024) and Jenkyn-Bedford et al. (2021), suggest that despite a network of interactions, the Pol epsilon catalytic domain likely occupies different positions during PCNA loading and processive leading strand synthesis. I have no further comments and congratulate the authors on their work.

We are delighted with the referee's positive assessment of our manuscript.

Dr. Joseph TP Yeeles
MRC Laboratory of Molecular Biology (LMB)
Francis Crick Avenue
Cambridge CB2 0QH
United Kingdom

4th Feb 2025

Re: EMBOJ-2024-119352R
Competition for the nascent leading strand shapes the requirements for PCNA loading in the replisome

Dear Joe,

Thank you for submitting your final revised manuscript for our consideration. I am pleased to inform you that following positive re-review by the original referee 2 (see comment below), we have now accepted it for publication in The EMBO Journal.

With kind regards,

Hartmut

Referee #2:

The authors have satisfactorily addressed all questions raised during the prior review.